# The HIV-1 Tat protein recruits a ubiquitin ligase to reorganize the 7SK snRNP for transcriptional activation

Tyler B Faust[1], Yang Li[1], Curtis W Bacon[2], Gwendolyn M Jang[3,4], Amit Weiss[1], Bhargavi Jayaraman[1], Billy W Newton[3,4], Nevan J Krogan[3,4], Iván D'Orso[2], Alan D Frankel[1]*

[1]Department of Biochemistry and Biophysics, University of California, San Francisco, San Francisco, United States; [2]Department of Microbiology, University of Texas Southwestern Medical Center, Dallas, United States; [3]Department of Cellular and Molecular Pharmacology, University of California, San Francisco, San Francisco, United States; [4]J David Gladstone Institutes, San Francisco, United States

**Abstract** The HIV-1 Tat protein hijacks P-TEFb kinase to activate paused RNA polymerase II (RNAP II) at the viral promoter. Tat binds additional host factors, but it is unclear how they regulate RNAP II elongation. Here, we identify the cytoplasmic ubiquitin ligase UBE2O as critical for Tat transcriptional activity. Tat hijacks UBE2O to ubiquitinate the P-TEFb kinase inhibitor HEXIM1 of the 7SK snRNP, a fraction of which also resides in the cytoplasm bound to P-TEFb. HEXIM1 ubiquitination sequesters it in the cytoplasm and releases P-TEFb from the inhibitory 7SK complex. Free P-TEFb then becomes enriched in chromatin, a process that is also stimulated by treating cells with a CDK9 inhibitor. Finally, we demonstrate that UBE2O is critical for P-TEFb recruitment to the HIV-1 promoter. Together, the data support a unique model of elongation control where non-degradative ubiquitination of nuclear and cytoplasmic 7SK snRNP pools increases P-TEFb levels for transcriptional activation.
DOI: https://doi.org/10.7554/eLife.31879.001

*For correspondence:
frankel@cgl.ucsf.edu

Competing interests: The authors declare that no competing interests exist.

## Introduction

The advent of genome-wide methods to interrogate transcription has shown that many genes in higher eukaryotes contain stably paused RNAP II molecules immediately downstream of transcription start sites (*Core et al., 2008*; *Henriques et al., 2013*; *Kwak and Lis, 2013*; *Rahl et al., 2010*; *Zeitlinger et al., 2007*). The release of RNAP II into productive elongation is controlled by the kinase positive transcription elongation factor b (P-TEFb), composed of cyclin T1 (CCNT1) and cyclin-dependent kinase 9 (CDK9) subunits (*Kwak and Lis, 2013*; *Marshall and Price, 1995*). P-TEFb hyper-phosphorylates DRB sensitivity inducting factor (DSIF), negative elongation factor (NELF), and the C-terminal domain of RNAP II, resulting in a processive polymerase (*Czudnochowski et al., 2012*; *Yamaguchi et al., 1999*). Numerous cellular proteins interact with P-TEFb to relieve pausing, including bromodomain-containing protein 4 (BRD4) (*Yang et al., 2005*), Mediator (*Takahashi et al., 2011*), nuclear factor (NF)-κB (*Barboric et al., 2001*), and c-Myc (*Eberhardy and Farnham, 2002*). The human immunodeficiency virus-1 (HIV-1) transcription factor, Tat, also interacts with P-TEFb to activate a paused polymerase at the viral promoter (*Feinberg et al., 1991*; *Wei et al., 1998*).

P-TEFb activity is controlled by sequestration into the inhibitory 7SK small nuclear ribonucleoprotein (snRNP) complex (*Nguyen et al., 2001*; *Yang et al., 2001*). In this complex, hexamethylene bisacetamide inducible 1 (HEXIM1) inhibits the kinase activity of CDK9 in a 7SK snRNA-dependent manner (*Michels et al., 2004*), while the 7SK snRNA methyl phosphate capping enzyme (MEPCE)

and La-related protein 7 (LARP7) stabilize the 7SK snRNA by adding a 5' methylphosphate cap or binding to the 3' poly(U) tail of the RNA, respectively (*He et al., 2008*). Multiple signaling pathways regulate the association of P-TEFb with the 7SK snRNP, tuning the required levels of active kinase in the cell (*Chen et al., 2008*; *Liu et al., 2014*). We have previously shown that 7SK snRNP-inhibited P-TEFb is recruited to the HIV-1 promoter (*D'Orso and Frankel, 2010*), where Tat activates transcription elongation by displacing the inhibitory subunits through binding of the nascently transcribed transactivation response element (TAR) RNA. Supporting this model of activation, recent genome-wide studies have revealed that snRNP-bound P-TEFb occupies the majority of promoters that display RNAP II pausing (*Ji et al., 2013*; *McNamara et al., 2016*). In this context, the splicing protein SC35 appears to act analogously to Tat by releasing the 7SK snRNP in a nascent RNA-binding-dependent step (*Ji et al., 2013*).

Ubiquitination is a post-translational modification in which the 76 amino acid protein ubiquitin becomes covalently attached via its C-terminus to a primary amine on a substrate protein. Beyond its well-known function as a signal for degradation, ubiquitination has emerged as a major signaling pathway in the cell, especially in transcription. As an example, mono-ubiquitination of histone H2B is associated with transcriptionally active chromatin and regulates subsequent histone modifications (*Hammond-Martel et al., 2012*). Ubiquitination of other transcription factors, including p53 and Myc, can lead to proteasomal degradation (*Haupt et al., 1997*; *Kim et al., 2003*), stabilization (*Popov et al., 2010*), or re-localization (*Le Cam et al., 2006*), underscoring the diversity of regulatory mechanisms for ubiquitination in transcriptional control.

Here, we report the identification of ubiquitin conjugating enzyme E2 O (UBE2O), a hybrid E2-E3 ubiquitin ligase (*Mashtalir et al., 2014*; *Vila et al., 2017*; *Zhang et al., 2013a*), as being critical for Tat transcriptional activity. We show that Tat recruits UBE2O to robustly ubiquitinate HEXIM1 in a non-degradative manner. Interestingly, UBE2O-dependent ubiquitinated HEXIM1 species are largely cytoplasmic, which led to the discovery of a cytoplasmic pool of P-TEFb-7SK snRNP. Tat-dependent HEXIM1 ubiquitination by UBE2O releases P-TEFb from this cytoplasmic reservoir with the 7SK snRNP, resulting in the nuclear import of the active kinase. Our results reveal an unanticipated amount of cytoplasmic regulation of HIV-1 transcription, where the Tat protein co-opts ubiquitination machinery to reorganize a major transcription elongation complex. Further, UBE2O activity is essential for the Tat-dependent recruitment of P-TEFb to the viral promoter, demonstrating the importance of this ubiquitin ligase in targeting P-TEFb to chromatin. Finally, this mechanism of transcription elongation control appears to be more broadly used as treatment of cells with the CDK9 inhibitor, DRB, similarly resulted in P-TEFb nuclear import and chromatin retention with an increased association between UBE2O and HEXIM1.

## Results

### UBE2O is a critical regulator of Tat transcriptional activity

Our previous proteomic interaction study identified several high-confidence host binding partners for the HIV-1 Tat protein (*Jäger et al., 2011b*), including well-established and novel interactions. While the contributions of the P-TEFb-based super elongation complex (SEC) (*He et al., 2010*; *Lin et al., 2010*; *Schulze-Gahmen et al., 2014*; *Sobhian et al., 2010*) and 7SK snRNP (*D'Orso and Frankel, 2010*; *Muniz et al., 2010*) to regulating viral transcription has become better defined, the functions of additional Tat interactors are largely unexplored. Many of these Tat factors are involved in ubiquitin signaling, including PJA2, a ubiquitin ligase that directly modifies Tat to regulate viral transcription (*Faust et al., 2017*). In addition, Tat also interacts with the ubiquitin ligases ZFP91 and UBE2O. ZFP91 ubiquitinates NF-κB-inducing kinase via a Lys63 linkage to activate a non-canonical NF-κB pathway (*Jin et al., 2010*). UBE2O is a hybrid E2-E3 ubiquitin ligase that can ubiquitinate multiple chromatin-associated proteins within their nuclear localization signal (NLS), resulting in cytoplasmic retention (*Mashtalir et al., 2014*). UBE2O has also recently been shown to induce the degradation of the mixed-lineage leukemia (MLL) protein, which is commonly fused to members of the P-TEFb-based super elongation complex in leukemia (*Liang et al., 2017*).

To explore the activity of ZFP91 and UBE2O in HIV-1 transcription, we performed RNAi-mediated knockdowns of these ubiquitin ligases coupled to a functional assay for Tat activity. We used a HeLa cell line harboring a single, full-length HIV-1 provirus deleted for the *tat* gene (HeLa[proviruSΔtat])

(*Faust et al., 2017*), which allowed us to precisely control transcriptional activation by transfection of a Tat plasmid and to monitor Tat activity in a relatively natural proviral context by the production of intracellular HIV-1 capsid protein (p24). A small quantity of Tat plasmid (0.5 ng) was transfected within the linear range of the activation assay approximating physiological Tat concentrations, and loss-of-function effects for the ligases were assessed (*Figure 1A*).

Multiple independent siRNAs targeting ZFP91 or UBE2O inhibited Tat activity by 2-fold or greater, an effect similar to knockdown of the P-TEFb subunits CCNT1 and CDK9 (*Figure 1A*, *Figure 1—figure supplement 1A*, *Figure 1—source data 1*). Importantly, both ligases are expressed in CD4$^+$ T cells (*Figure 1—figure supplement 1B*), underscoring that they can regulate Tat activity during normal viral infection of target immune cells. To determine whether ZFP91 and UBE2O specifically regulate transcription elongation, we measured promoter-proximal and -distal HIV-1 RNA production in the HeLa$^{provirusΔtat}$ cells and found that UBE2O knockdown significantly reduced the Tat-dependent production of distal, but not proximal, transcripts (*Figure 1B*, *Figure 1—source data 2*). These data strongly implicate UBE2O as an essential regulator of HIV-1 transcription elongation.

## UBE2O interacts with the HEXIM1 protein of the 7SK snRNP

Since the 7SK snRNP complex is critical in regulating pause release at the HIV-1 promoter (*D'Orso and Frankel, 2010*), we affinity purified the HEXIM1 protein of the snRNP for mass spectrometry to better define the machinery involved in pause release and potentially make connections to the ubiquitin ligases. As expected, we observed interactions with P-TEFb and other components of the snRNP complex (*Figure 1C*). Intriguingly, UBE2O was identified as a HEXIM1 interactor from multiple purifications (*Figure 1C*, *Supplementary file 1*). The HEXIM1-UBE2O interaction was also identified in a recent effort to map the human interactome by systematic AP-MS of 2594 bait proteins (*Huttlin et al., 2015*), further validating the interaction. As additional support for the interaction of HEXIM1 and UBE2O, both proteins were identified in peak overlapping fractions from gel filtration of whole cell extracts (*Figure 1D*). As both HEXIM1 and Tat can bind UBE2O, we also examined how Tat expression might alter the HEXIM1-UBE2O interaction. Tat dramatically increased the association between endogenous HEXIM1 and transfected UBE2O (*Figure 1E*), supporting the ability of a viral transcription factor to induce the binding between a ubiquitin ligase and a major transcriptional regulatory protein. We next performed a tandem affinity purification of Tat-STREP and UBE2O-FLAG followed by western blotting of endogenous HEXIM1, which demonstrated that these three proteins can form a multi-protein complex (*Figure 1F*). Given these strong connections to HEXIM1 and because the 7SK snRNP plays a key role in activating viral transcription elongation, we decided to further investigate the activity of UBE2O in regulating Tat function.

## Tat uses a novel interaction surface to bind the 7SK snRNP and UBE2O

Tat forms an extensive interaction surface on the CCNT1 subunit of P-TEFb (*Tahirov et al., 2010*). Given that Tat also potently recruits UBE2O to HEXIM1 (*Figure 1E and F*), we reasoned that Tat likely uses an additional binding surface to promote this interaction. To explore whether a novel functional interface might exist, we used a data set in which all Tat alanine point mutants were tested for activity in a transcriptional reporter system (*D'Orso et al., 2012*; *Fernandes et al., 2016*) and mapped the activities of these mutants onto the Tat–P-TEFb crystal structure (*Tahirov et al., 2010*). Several known regions in Tat were clearly defined where mutants displayed low transcriptional activity, including the cysteine-rich zinc-binding clusters (*Figure 2A*) and a stretch of hydrophobic residues (Phe38, Leu43, Gly44, and Ile45) buried in CCNT1 (*Figure 2B*). In support of a distinct interaction surface, a cluster of highly conserved, surface-exposed residues (Glu2, Tyr26, Lys28, and Phe32) displayed severe loss-of-function phenotypes and faced away from CCNT1 and CDK9 (*Figure 2C*).

To directly test if these residues constituted a new binding surface, we purified wild type and mutant Tat proteins from transiently transfected HEK 293 T cells and examined the interactions with endogenous proteins. Indeed, Tat K28A poorly bound to UBE2O and the 7SK snRNP proteins MEPCE and LARP7 (*Figure 2D*, *Figure 2—figure supplement 1* for inputs). Of note, the interactions of these snRNP proteins with Tat is entirely RNA-dependent (*Figure 2D*, RNAse A treatment), implicating Lys28 in 7SK snRNA binding in addition to its role in modulating the TAR RNA interaction

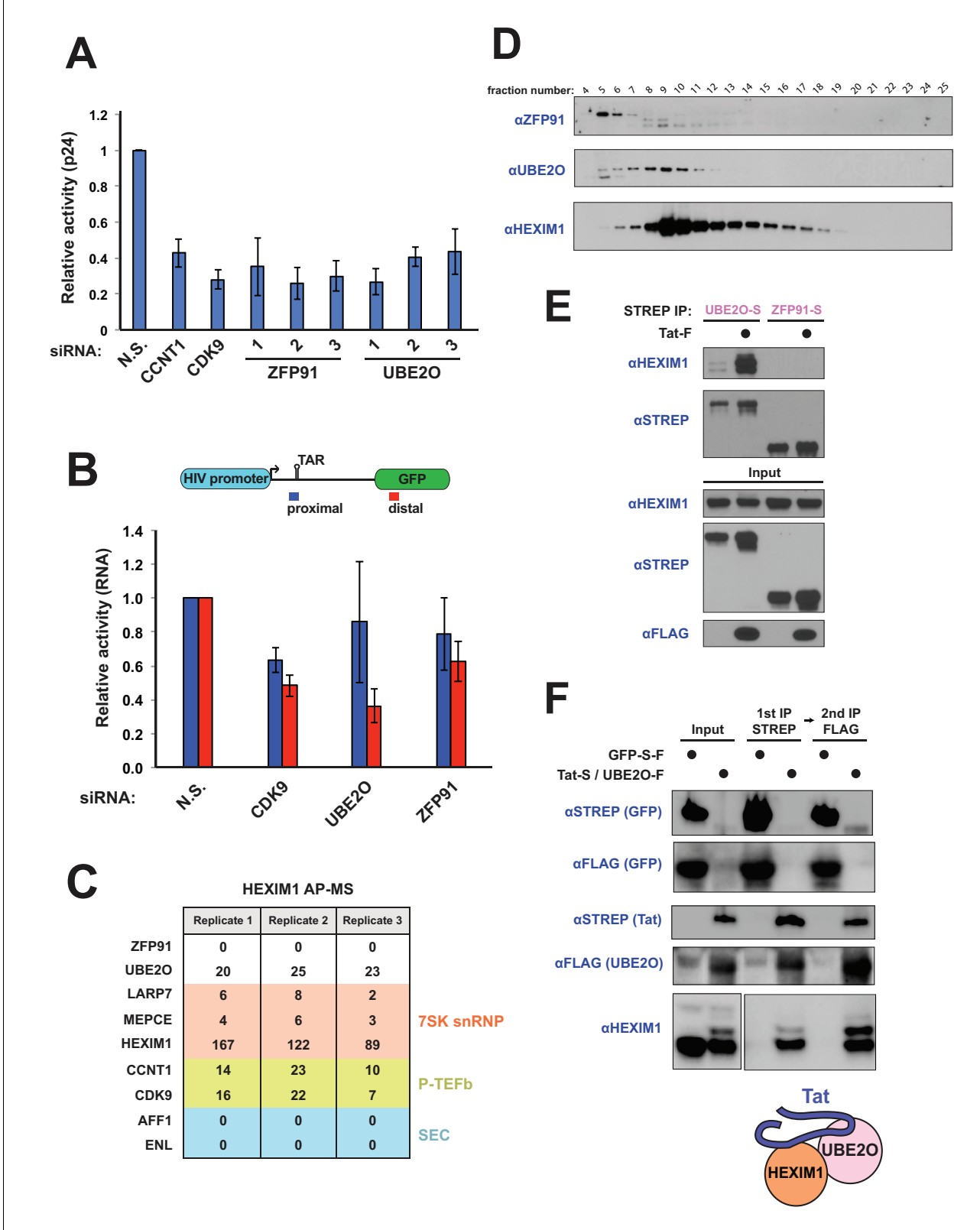

**Figure 1.** UBE2O is a critical regulator of Tat-dependent transcription and binds HEXIM1 of the 7SK snRNP. (**A**) UBE2O and ZFP91 RNAi knockdown inhibit Tat-dependent transcription. Normalized, relative Tat activities ([p24$^{host\ RNAi}$/FFL$^{host\ RNAi}$] / [p24$^{N.S.\ RNAi}$/FFL$^{N.S.\ RNAi}$]) for CCNT1, CDK9, UBE2O, and ZFP91 are shown, and data are represented as the mean ± SEM of at least three biological replicates (defined as independent transfections. Cells were transfected independently on at least 2 different days) for independent siRNA transfections in the HeLa$^{provirusΔtat}$ cell line. (**B**) HIV RNA elongation

*Figure 1 continued on next page*

*Figure 1 continued*

assay with ZFP91 (si2), UBE2O (si1), or CDK9 RNAi. Knockdowns were performed in the same cells as (**A**), which contain GFP cloned into the Nef locus. qPCR was used to monitor the Tat-dependent production of proximal (blue) and distal (red) RNA transcripts, with the relative positions of the primer sets shown above. Data are presented as the mean ± SEM of relative Tat activity of at least three biological replicates (same definition as above). (**C**) AP-MS results using HEXIM1-FLAG as the bait protein. Peptide counts are shown for selected prey proteins from triplicate biological purifications (as defined above). See also **Supplementary file 1** for a complete list of interacting proteins. (**D**) Gel filtration of HEK 293T whole cell lysate. Fractions were collected after the void volume and Western blotted for the indicated proteins. ZFP91 is shown as a specificity control. (**E**) STREP purification of transfected proteins ±Tat co-transfection. The endogenous HEXIM1 interaction was monitored by western blot using an anti-HEXIM1 antibody. (**F**) Tandem affinity purification of Tat-STREP and UBE2O-FLAG from transfected cells. GFP-S-F (which is STREP and FLAG tagged) was used as a negative control in the experiment. The endogenous HEXIM1 interaction was detected by antibody. Different exposure times from the same blot for the HEXIM1 input and elutions are shown for ease of comparison. Schematic illustrates the Tat-HEXIM1-UBE2O complex.

DOI: https://doi.org/10.7554/eLife.31879.002

The following source data and figure supplement are available for figure 1:

**Source data 1.** Source data for luciferase-normalized p24 values of siRNA knockdown coupled to Tat activity of **Figure 1A**.
DOI: https://doi.org/10.7554/eLife.31879.004
**Source data 2.** Source data for proximal and distal Tat-dependent RNA species of **Figure 1B**.
DOI: https://doi.org/10.7554/eLife.31879.005
**Figure supplement 1.** RNAi-mediated protein knockdown and endogenous expression.
DOI: https://doi.org/10.7554/eLife.31879.003

(**D'Orso and Frankel, 2009**). Strikingly, mutation of Phe32 (F32A), which is adjacent to Lys28 in the structure, significantly increased binding to UBE2O by greater than 4-fold. This might indicate that Phe32 is involved in a handoff from UBE2O to another protein, such that its mutation locks Tat in the UBE2O-bound state and completely inhibits transcriptional activity. Importantly, the P-TEFb interaction was unaffected by Lys28 or Phe32 mutations (**Figure 2D**), unlike the mutation of Phe38 (F38A) and Leu43 (L43A) residues that are buried in CCNT1 and disrupt P-TEFb binding, corroborating that Lys28 and Phe32 comprise a novel docking surface.

The SEC scaffold protein, AFF1, had an identical binding profile to CDK9 and CCNT1, supporting recent work that AFF1 is a ubiquitous P-TEFb binding partner (**Lu et al., 2014**). Transcriptional activity (**D'Orso et al., 2012**; **Fernandes et al., 2016**) and binding data both support a model in which Tat uses a buried hydrophobic patch (Phe38, Leu43) to anchor onto P-TEFb while using two spatially adjacent residues (Lys28, Phe32) to stabilize interactions with the 7SK snRNP and UBE2O (**Figure 2E**). Further, residues Lys28 and Phe32 were highly selected for during viral replication as assessed by deep mutational scanning, supporting a critical role of these amino acids (**Fernandes et al., 2016**). Thus, Tat encodes an essential hijacking function directed at a ubiquitin ligase and the 7SK snRNP.

## Tat hijacks UBE2O to mono-ubiquitinate HEXIM1 in a non-degradative manner

Given that Tat induces the association of UBE2O and HEXIM1 (**Figure 1E**) and encodes a novel interaction surface for UBE2O and the 7SK snRNP (**Figure 2D**), we wanted to determine whether UBE2O ubiquitinates HEXIM1 and potentially other snRNP proteins, and how Tat might regulate this process. To test this, we first pulled down 7SK snRNP complex subunits and, upon blotting for transfected HA-tagged ubiquitin, found that UBE2O co-expression dramatically increased HEXIM1 and MEPCE ubiquitination but not LARP7, CCNT1, or CDK9 (**Figure 3A**). Correlating with the ubiquitination results, AP-MS experiments with the LARP7 and MEPCE subunits identified MEPCE, but not LARP7, as interacting with UBE2O (**Figure 3—figure supplement 1A,B**). As HEXIM1 is the key regulatory kinase inhibitor of the P-TEFb kinase (**Michels et al., 2004**) and is known to be ubiquitinated (**Lau et al., 2009**), we further characterized the HEXIM1 modification.

We first investigated the nature of the UBE2O-dependent ubiquitin linkage on HEXIM1, as recent work has shown that UBE2O monoubiquitinates numerous proteins (**Mashtalir et al., 2014**; **Nguyen et al., 2017**; **Yanagitani et al., 2017**; **Zhang et al., 2013a**). Similar to previous studies, expression of a lysine-free ubiquitin mutant (Ub ΔK), which blocks the ability to form polyubiquitin chains, did not affect UBE2O-dependent HEXIM1 ubiquitination in vivo (**Figure 3B**), demonstrating that HEXIM1 is monoubiquitinated at multiple sites. Further, a catalytically inactive mutant of UBE2O

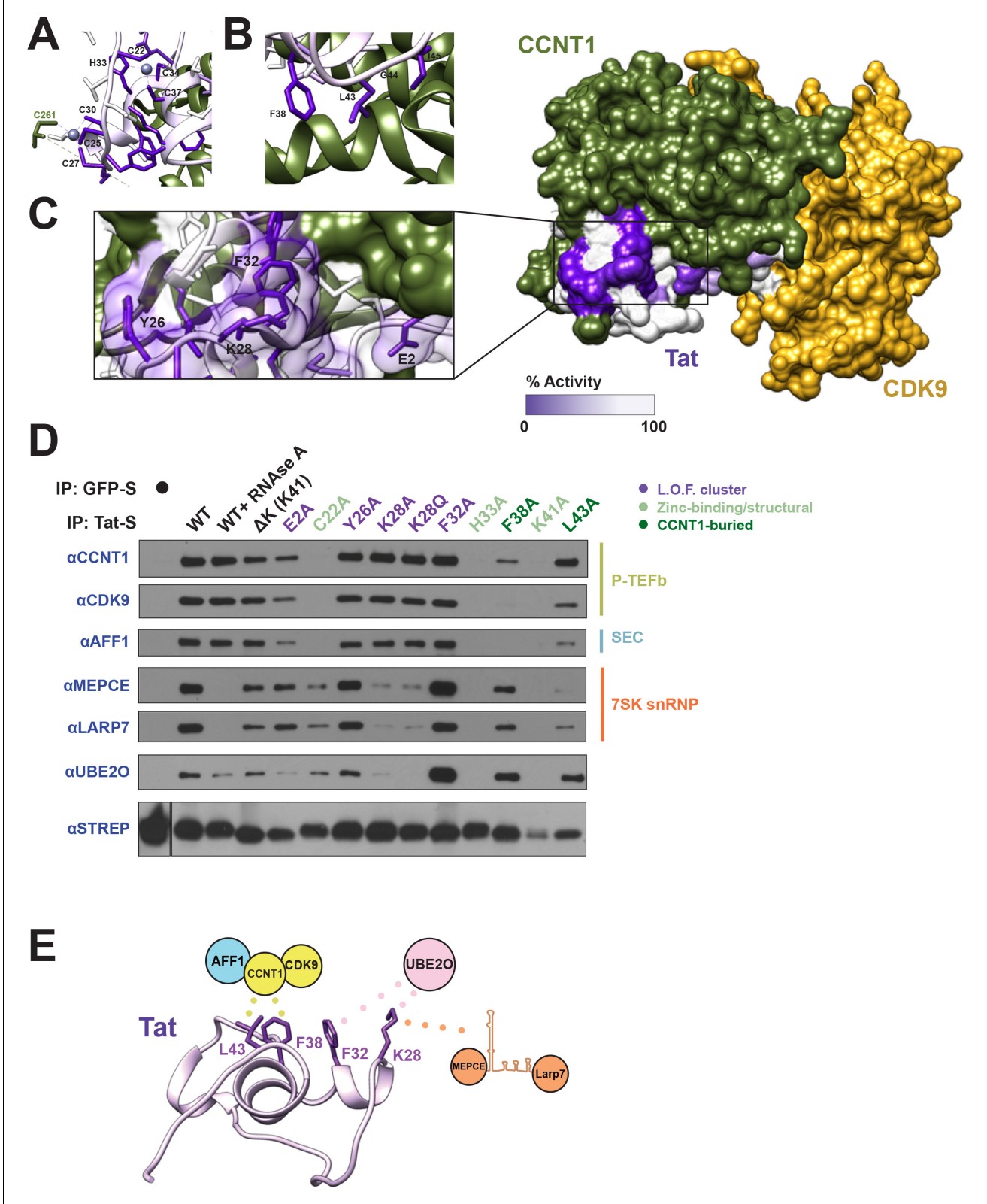

**Figure 2.** Tat uses a novel interaction surface to interact with the 7SK snRNP and UBE2O. (A) Tat transcriptional activity data collected from our systematic set of alanine point mutants (*D'Orso et al., 2012*; *Fernandes et al., 2016*) was mapped onto the crystal structure of the Tat-P-TEFb complex (PDB i.d. 3MI9). The most severe loss-of-function (L.O.F.) mutations are shaded in deep purple while those with no effect on activity are shaded in white. The cysteine-rich zinc-binding clusters (A) and a hydrophobic stretch buried in CCNT1 (B) are critically important for Tat function. (C) A

*Figure 2 continued on next page*

Figure 2 continued

surface-exposed patch of severe loss-of-function mutants (Glu2, Tyr26, Lys28, and Phe32) does not contact CCNT1 or CDK9. (D) IP of transiently transfected wild type or mutant STREP-tagged Tat proteins. GFP-S was used as a negative control. Western blots were performed on STREP elutions using antibodies to the endogenous proteins indicated. Important residues are color coded as indicated. L.O.F. cluster is the four surface-exposed mutants from (C). See also *Figure 2—figure supplement 1* for inputs. (E) Model of Tat interaction surfaces. Tat uses a hydrophobic patch including Leu43 and Phe38 buried at the CCNT1 interface, which also results in the binding of CDK9 and AFF1. Mutation of Lys28 reduces the interactions with both the 7SK snRNP proteins and UBE2O, whereas mutation of Phe32 dramatically increases the UBE2O interaction, associating the ligase with this residue.

DOI: https://doi.org/10.7554/eLife.31879.006

The following figure supplement is available for figure 2:

**Figure supplement 1.** Protein input blots for Tat mutant IPs.

DOI: https://doi.org/10.7554/eLife.31879.007

in the ubiquitin-conjugating domain (C1040S or CD) did not ubiquitinate HEXIM1 (*Figure 3—figure supplement 2A*), supporting specific modification by UBE2O. We next reconstituted ubiquitination of HEXIM1 by UBE2O in vitro and found that, like our observations in vivo, wild-type UBE2O but not the CD mutant robustly modified HEXIM1 (*Figure 3C*, *Figure 3—figure supplement 2B*). As also observed in vivo, multiple high-molecular-weight HEXIM1 species were generated using the Ub ΔK variant. Collectively, these results reveal that HEXIM1 is a UBE2O substrate that is monoubiquitinated at multiple sites.

Although targeting proteins for proteasomal destruction is a well-known function of ubiquitination, it is increasingly appreciated that this modification, especially mono-ubiquitination, also serves important signaling functions. Indeed, UBE2O-dependent ubiquitination of HEXIM1 was unaffected by treatment with the proteasome inhibitor MG132 (*Figure 3D*, *Figure 3—figure supplement 2C* showing MG132 control, *Figure 3—figure supplement 2D* showing extent of UBE2O overexpression), supporting a non-degradative mechanism.

Strikingly, Tat expression increased HEXIM1 non-degradative mono-ubiquitination (*Figure 3D*), supporting a model of Tat-mediated HEXIM1 ubiquitination by UBE2O. Tat expression also increased the ubiquitination of endogenous HEXIM1, but not CCNT1 (*Figure 3—figure supplement 3A*), supporting the specificity of UBE2O-dependent ubiquitination as observed in transfected cells (*Figure 3A*). Furthermore, endogenous HEXIM1 that is associated with Tat, likely as part of the Tat-UBE2O-HEXIM1 complex (*Figure 1F*), is in a modified state (*Figure 3—figure supplement 3B*), including a species with the same molecular weight as mono-ubiquitinated HEXIM1 (refer to *Figure 3A*). Importantly, Tat-stimulated non-degradative ubiquitination of endogenous HEXIM1 (*Figure 3E*) was completely blocked by siRNA-mediated knockdown of UBE2O (*Figure 3E*), signifying that Tat hijacks the enzyme for HEXIM1 mono-ubiquitination. In addition, UBE2O knockdown prevented endogenous, Tat-independent HEXIM1 ubiquitination, but did not increase HEXIM1 protein levels (*Figure 3E*), further confirming that HEXIM1 is an endogenous substrate for non-degradative UBE2O mono-ubiquitination, and is not an artifact of UBE2O overexpression. Importantly, HEXIM1 ubiquitination by UBE2O is a function that is exploited and induced by Tat (*Figure 3E*) to modulate viral transcription.

## UBE2O ubiquitinates the NLS of HEXIM1

HEXIM1 ubiquitination resembles the modification of a recently described class of UBE2O substrates that are transcription factors with acceptor lysines contained within a bipartite NLS surrounding a patch of aliphatic hydrophobic amino acids (*Mashtalir et al., 2014*), termed the 'VLI patch', which is also found in UBE2O itself and results in auto-ubiquitination. This bipartite NLS motif is consistent with UBE2O's preference for juxtaposed hydrophobic and basic patches (*Yanagitani et al., 2017*). Interestingly, while not identical, HEXIM1 has a similar motif (*Figure 4A*), with an NLS composed of 12 and 3 basic amino acid segments surrounding a stretch of mostly aromatic hydrophobic (W, Y) amino acids. To identify the sites of ubiquitination in HEXIM1 and to determine if the NLS is indeed a target, we first generated a fully lysine-free HEXIM1 mutant (ΔK) in which all lysines were mutated to arginine and observed no modification (*Figure 4B*, last lane), demonstrating that ubiquitination is lysine-dependent. When all lysines within the N- and C-termini were mutated to arginines (CTΔK + NTΔK), leaving lysines intact only within the NLS, HEXIM1 was still ubiquitinated

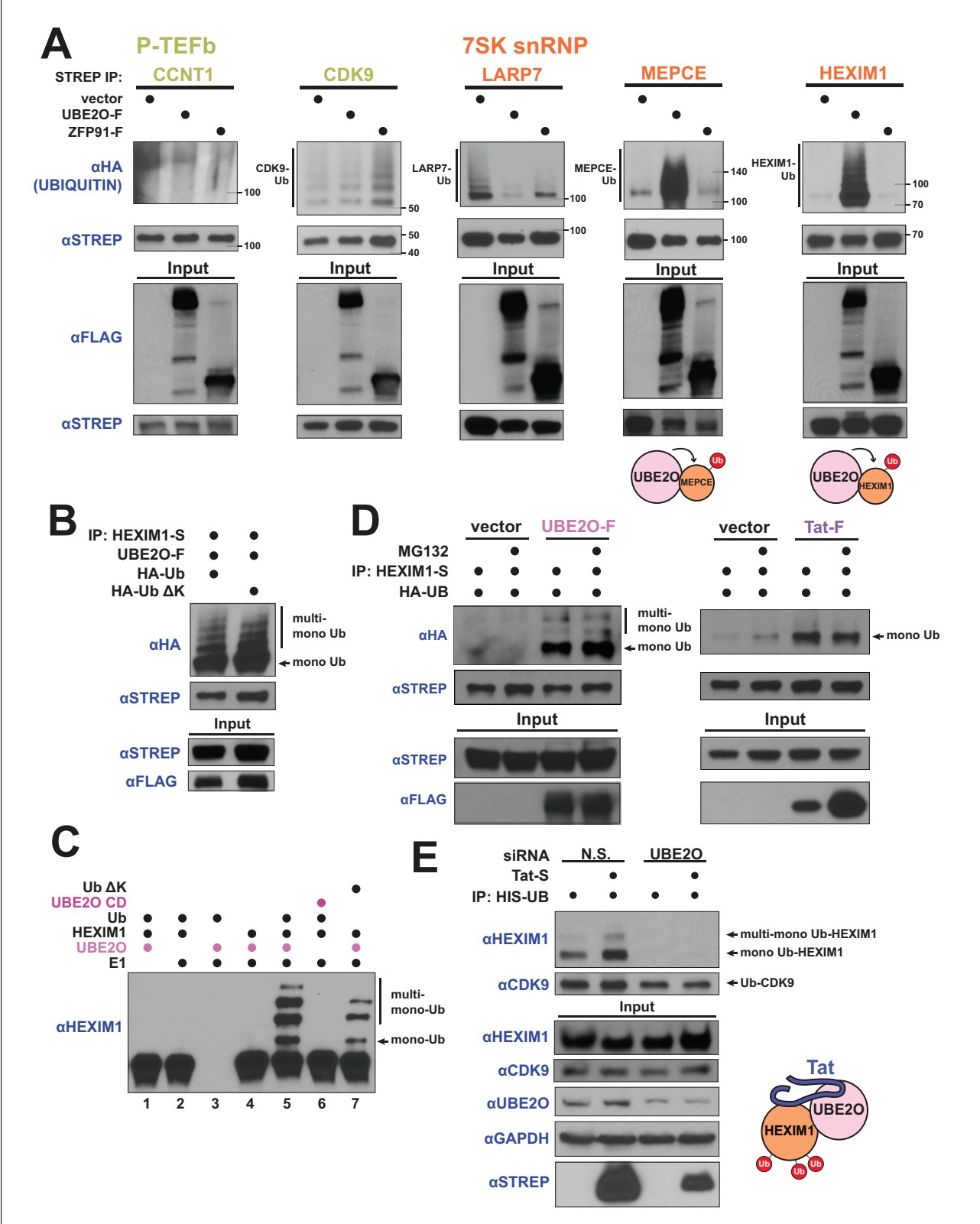

**Figure 3.** HIV-1 Tat hijacks UBE2O for the non-degradative monoubiquitination of HEXIM1. (**A**) In vivo ubiquitination of transfected, STREP-tagged P-TEFb and 7SK snRNP proteins co-transfected with HA-ubiquitin ±co transfection of UBE2O-F or ZFP91-F (-F designates FLAG tagged protein). ZFP91 is used as a specificity control. Denaturing lysis and IPs were performed to detect ubiquitination by HA western blot. Ubiquitinated species are indicated. Schematics illustrate that UBE2O increases ubiquitination of the 7SK snRNP proteins MEPCE and HEXIM1. (**B**) In vivo ubiquitination of

*Figure 3 continued on next page*

Figure 3 continued

HEXIM1-S in HEK 293 T cells co-transfected with UBE2O-F and either wild type or lysine-free (ΔK) HA-ubiquitin. Monoubiquitinated HEXIM1 and additional ubiquitinated species are indicated. (C) In vitro ubiquitination of HEXIM1 by UBE2O. HEXIM1-S purified from HEK 293T cells was incubated with recombinant E1 and ubiquitin, UBE2O-F purified from HEK 293T cells, and ATP. Control reactions lacked individual components as indicated. Additionally, one reaction was incubated with UBE2O CD (catalytically dead) purified from HEK 293T cells in place of wild type UBE2O (lane 6) or recombinant ubiquitin ΔK in place of wild type ubiquitin (lane 7). See also *Figure 3—figure supplement 2B*. (D) In vivo ubiquitination of HEXIM1-S co-transfected with UBE2O-F or Tat-F and subsequently treated with DMSO control or MG132 to block proteasomal degradation. See also *Figure 3—figure supplement 2C*. (E) HEK 293 T cells were transfected with non-silencing or UBE2O-directed siRNA (si1). Knockdown cells were then transfected with HIS-Ubiquitin L73P followed by a denaturing nickel purification. The L73P ubiquitin mutant was used to enrich for ubiquitinated species, which were detected by western blot using specific antibodies (anti-CDK9, anti-HEXIM1) on the nickel-purified elution and are indicated. Schematic illustrates that Tat enhances HEXIM1 ubiquitination through UBE2O.

DOI: https://doi.org/10.7554/eLife.31879.008

The following figure supplements are available for figure 3:

**Figure supplement 1.** FLAG-tagged MEPCE and LARP7 AP-MS.

DOI: https://doi.org/10.7554/eLife.31879.009

**Figure supplement 2.** A catalytically inactive UBE2O does not induce HEXIM1 ubiquitination.

DOI: https://doi.org/10.7554/eLife.31879.010

**Figure supplement 3.** Tat induces the ubiquitination of endogenous HEXIM1 and MEPCE.

DOI: https://doi.org/10.7554/eLife.31879.011

(*Figure 4B*), consistent with acceptor residues being located in the NLS. Several additional mutants suggest that lysines within the C-terminus can also be ubiquitinated (see *Figure 4B* legend), but we focused on the NLS modifications because of the previously defined connections to UBE2O (*Mashtalir et al., 2014*).

To more precisely map the acceptor sites, we used mass spectrometry (MS) and identified multiple ubiquitinated lysines in the NLS and C-terminus (*Figure 4C*, *Supplementary file 2*), supporting our initial mapping (*Figure 4B*). Since the HEXIM1 NLS is lysine and arginine-rich, not all peptides covering this region are detectable by MS after trypsin or Arg-C digestion. Therefore, as an alternative approach to map acceptor sites in the NLS, we restored individual lysines within the lysine-free (ΔK) background and found that four (Lys150, Lys151, Lys152, and Lys159) recovered monoubiquitination, with Lys151 and Lys159 being the most robust (*Figure 4D*). These four lysines are located within the first segment of the bipartite NLS, suggesting that they are the most accessible for UBE2O ubiquitination.

One intriguing possibility is that HEXIM1 is modified by UBE2O in the context of the 7SK snRNP, especially given the observation that Tat also stimulates HEXIM1 modification (*Figure 3D,E*) and is known to be in complex with the 7SK snRNP (*D'Orso and Frankel, 2010*; *Jäger et al., 2011b*; *Muniz et al., 2010*; *Sobhian et al., 2010*). Modification within the 7SK snRNP is also suggested by UBE2O binding and ubiquitination of MEPCE (*Figure 3A*, *Figure 3—figure supplement 1A*). In support of this model, a HEXIM1 mutant deficient for 7SK RNA binding (150–156 Ala) (*Michels et al., 2004*; *Yik et al., 2004*) displayed a complete loss of UBE2O-dependent ubiquitination, in striking contrast to another HEXIM1 mutant deficient for P-TEFb binding that was still fully ubiquitinated ($_{202}PYNT_{205}$ -> $_{202}PDND_{205}$) (*Michels et al., 2004*) (*Figure 4E*). Because the 150–156 Ala mutant also lacks some of the lysines that become ubiquitinated (*Figure 4D*), we generated a HEXIM1 150–156 Arg mutant which similarly lacks the NLS acceptor lysines but retains RNA binding (*Lau et al., 2009*) and found that it was ubiquitinated equally to wild-type HEXIM1 (*Figure 4E*). Thus, RNA binding by HEXIM1 appears to be a prerequisite for UBE2O recognition and ubiquitination. 7SK RNA-binding releases HEXIM1 from an auto-inhibited state (*Barboric et al., 2005*), which likely establishes the proper conformation of HEXIM1 for UBE2O ubiquitination.

## UBE2O influences HEXIM1 subcellular localization

Given that UBE2O ubiquitinates the NLS of HEXIM1, we next asked whether this modification resulted in a redistribution of HEXIM1 subcellular localization. Expression of UBE2O, itself a cytoplasmic ubiquitin ligase, altered the localization of endogenous HEXIM1 with an increase of HEXIM1 in the cytoplasm and a simultaneous decrease in the nuclear pellet fraction containing chromatin (*Figure 5A*). Consistent with the cytoplasmic accumulation of HEXIM1, the majority of the UBE2O-

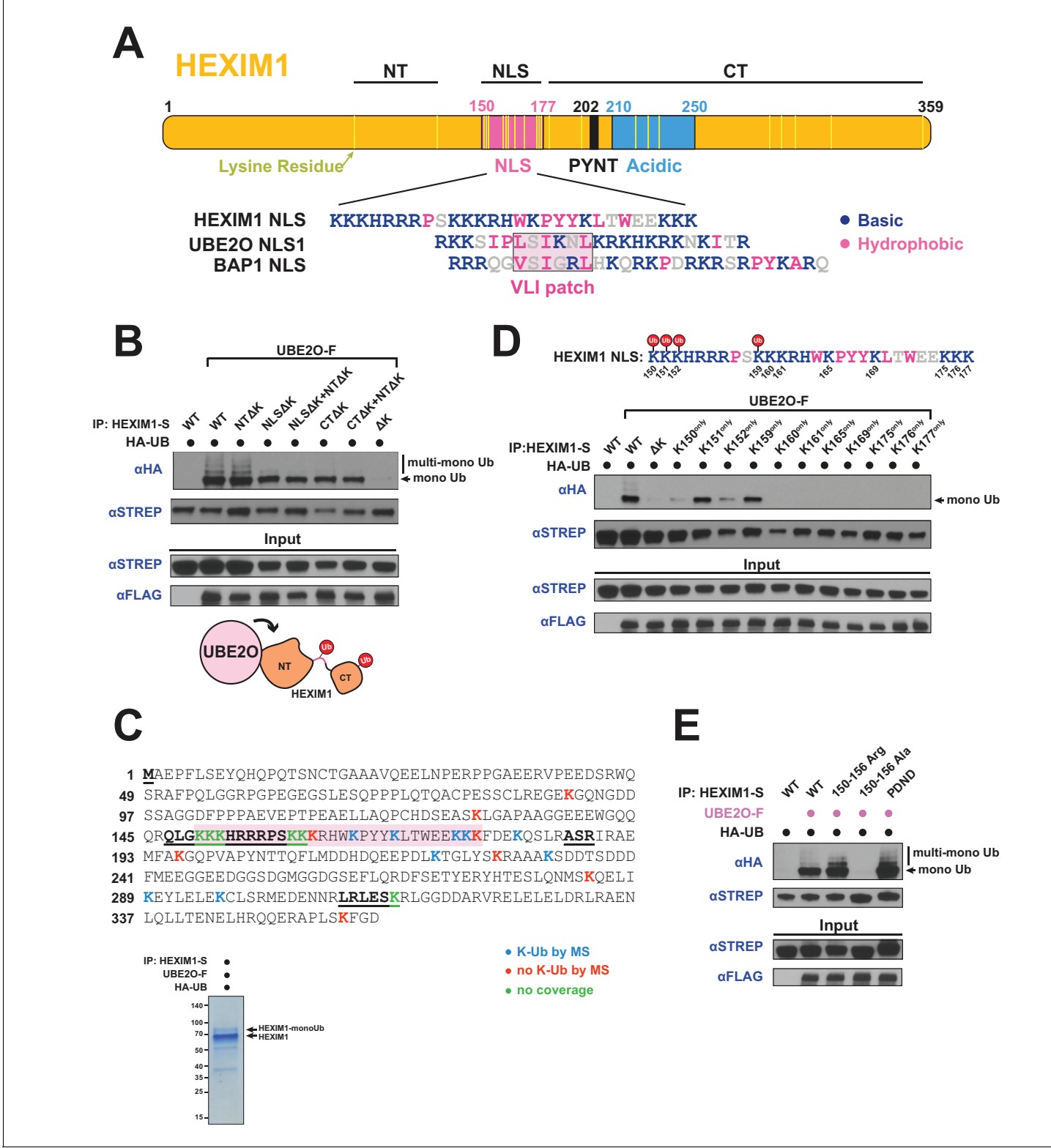

**Figure 4.** UBE2O ubiquitinates the NLS of HEXIM1. (**A**) Domain organization of HEXIM1. The positions of the 23 lysines in HEXIM1 are indicated by yellow bars in the C-terminal (CT) and N-terminal (NT) regions and the NLS. (Below) Alignment of the NLS sequences from HEXIM1, UBE2O, and a known UBE2O substrate, BAP1 (*Mashtalir et al., 2014*). UBE2O and several of its target proteins contain a consensus bipartite NLS architecture that shares similarities with the HEXIM1 NLS. Basic residues are labeled in blue while hydrophobic residues are labeled in pink. (**B**) In vivo ubiquitination of HEXIM1 lysine mutants. Combinations of STREP-tagged lysine-to-arginine mutants were generated in the HEXIM1 N-terminal region (NTΔK), NLS

*Figure 4 continued on next page*

Figure 4 continued

(NLSΔK), and C-terminal region (CTΔK). A complete lysine-free HEXIM1 is designated as ΔK. UBE2O-F was co-transfected where indicated, and ubiquitination was detected by anti-HA western after denaturing lysis and IP. Monoubiquitin and multi-monoubiquitin species are indicated. The NTΔK mutant had no effect on ubiquitination, suggesting that there are no acceptor sites in the N-terminal region. However, both the NLSΔK and CTΔK mutants displayed decreased ubiquitination, demonstrating the presence of acceptor lysines in these regions. Schematic illustrates that lysines in the NLS (pink connection between domains) and in the C-terminal region function as acceptor sites. See also **Supplementary file 2**. (**C**) Coverage of the HEXIM1 sequence by MS. Sequences not identified by MS are bolded and underlined. Shaded in red is the HEXIM1 bipartite NLS. Color coding reflects whether a lysine was identified with a ubiquitin di-glycine tryptic remnant fragment. Shown below is a protein gel of HEXIM1-S purified under denaturing conditions from HEK 293 T cells with the co-expression of UBE2O-F. Arrows indicate unmodified and monoubiquitinated HEXIM1 species. (**D**) In vivo ubiquitination of individual lysine residues in the HEXIM1 NLS re-introduced in the context of the lysine-free (ΔK) HEXIM1 background. Ubiquitination of single-lysine HEXIM1 proteins was detected by HA Western blot. (top) All lysines are numbered in the HEXIM1 NLS and those that restore ubiquitination are indicated. (**E**) In vivo ubiquitination of wild-type STREP-tagged HEXIM1 or mutants deficient in binding P-TEFb ($_{202}$PYNT$_{205}$ -> $_{202}$PDND$_{205}$) or the 7SK snRNA (150–156 Ala). HEXIM1-S was co-transfected with UBE2O-F as indicated.
DOI: https://doi.org/10.7554/eLife.31879.012

dependent ubiquitinated species of HEXIM1 were also cytoplasmic (**Figure 5B**, lane 2), suggesting that HEXIM1 may be ubiquitinated in this fraction to regulate its localization and inhibit chromatin recruitment. It is also feasible that a nuclear fraction of UBE2O ubiquitinates nuclear HEXIM1, leading to its redistribution to the cytoplasm. This is particularly intriguing given UBE2O's known nucleo-cytoplasmic shuttling activity and interactions with nuclear import/export machinery (**Mashtalir et al., 2014**). Interestingly, the lysine-free HEXIM1ΔK mutant, which is not ubiquitinated by UBE2O (**Figure 4B**), accumulated greater than threefold more in the chromatin-containing nuclear pellet compared to wild-type HEXIM1 when expressed in HEK 293 T cells (**Figure 5C**), further supporting a role of HEXIM1 lysine modification in regulating its localization. In this way, UBE2O likely acts to potentiate transcription by ubiquitinating HEXIM1 and promoting its sequestration in the cytoplasm, a mechanism co-opted by Tat during viral infection.

## HEXIM1 ubiquitination regulates interactions with the 7SK snRNP and is required for Tat-dependent P-TEFb release from the snRNP

In addition to ubiquitinating the HEXIM1 NLS, UBE2O appeared to first require RNA binding by HEXIM1 for ubiquitination (**Figure 4E**). Given this requirement, we explored whether UBE2O ubiquitination might alter protein-protein interactions within a HEXIM1 RNP complex (**Ray et al., 2014**; **Werner et al., 2015**), in particular the 7SK snRNP or the recently described HEXIM1-NEAT1 RNP (**Morchikh et al., 2017**). Interestingly, the knockdown of UBE2O resulted in an increased association of CCNT1, CDK9, and LARP7 with endogenously purified HEXIM1 (**Figure 6A**), with no obvious effect on the HEXIM1 interactions with proteins of the NEAT1 RNP. This result supports the idea that UBE2O regulates HEXIM1 activity as part of the 7SK snRNP. Further, while Tat expression released HEXIM1 from the 7SK snRNP, with no effect on NEAT1 RNP interactions (**Figure 6A**), knockdown of UBE2O limited the ability of Tat to disrupt HEXIM1-CCNT1 binding. Together, these results underscore that UBE2O is involved in specifically regulating HEXIM1's interaction with the 7SK snRNP in cells.

To further explore the role of UBE2O and Tat in altering HEXIM1's interactions with the 7SK snRNP, we utilized the transcriptionally inactive Tat F32A variant, which has a mutant interaction phenotype that is specific to UBE2O (**Figure 2D**) and would allow us to uncover the role of this ligase in P-TEFb release from the 7SK snRNP. Unlike wild-type Tat, the F32A mutant was unable to stimulate HEXIM1 ubiquitination (**Figure 6B**) despite its increased association with UBE2O (**Figure 2D**), further supporting a non-productive binding mode of this Tat mutant to the ligase. Further, the F32A mutant was deficient in releasing P-TEFb from endogenous 7SK snRNP-bound HEXIM1 compared to wild type Tat (**Figure 6C**), corroborating a role for HEXIM1 ubiquitination in the P-TEFb eviction step by Tat. To further support the role of HEXIM1 ubiquitination in the release of P-TEFb, we utilized the HEXIM1 ΔK mutant, which is not ubiquitinated by UBE2O (**Figure 4B**). This lysine-free mutant was refractory to complete P-TEFb release by Tat (**Figure 6D**) compared to wild-type HEXIM1. Altogether, the data firmly support that HEXIM1 ubiquitination by UBE2O is a critical step in the eviction of P-TEFb from the 7SK snRNP by Tat.

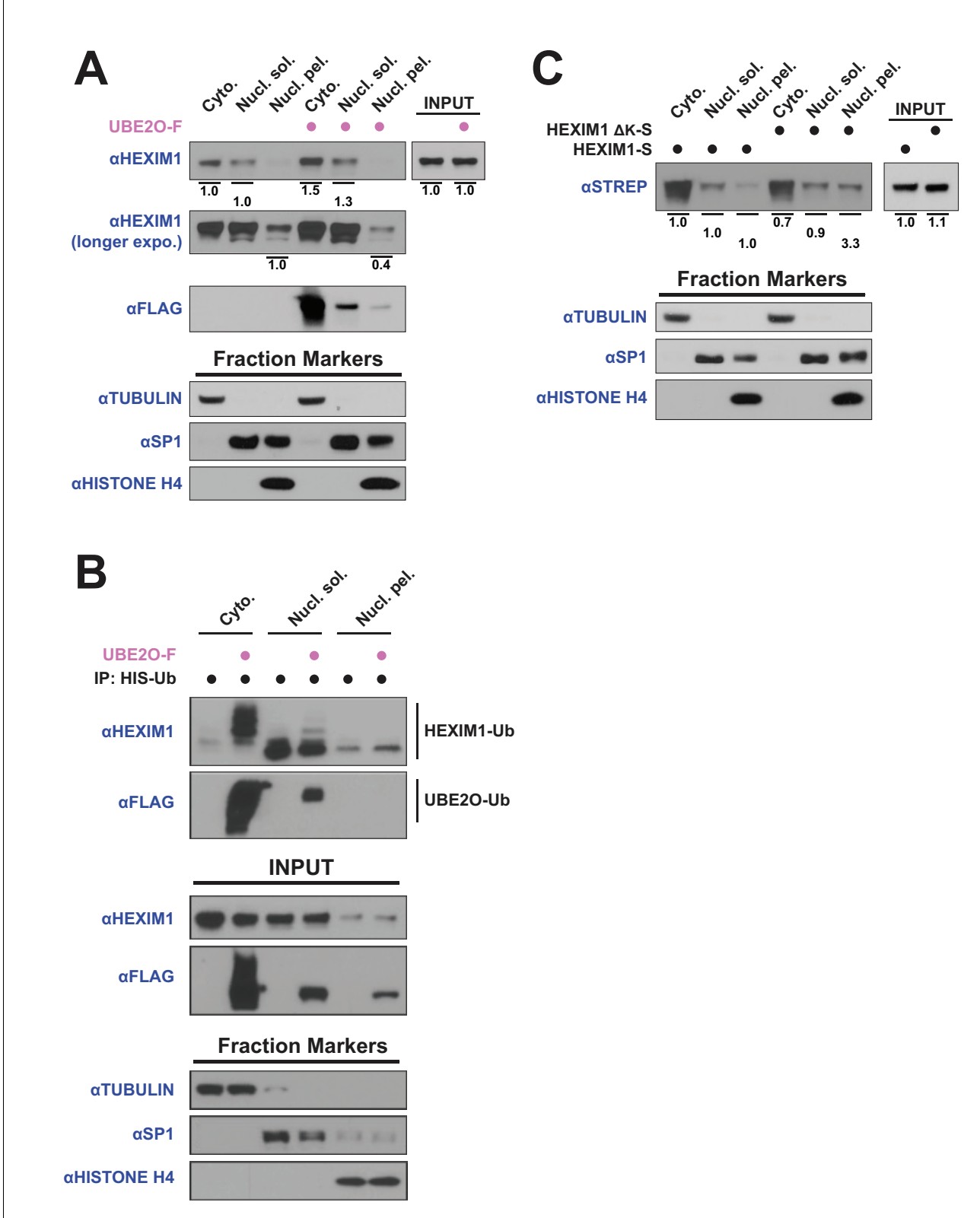

**Figure 5.** UBE2O influences HEXIM1 subcellular localization. (**A**) Subcellular localization of endogenous HEXIM1. HEK 293T were transfected ±3 μg UBE2O-F plasmid and fractionated into cytoplasmic (Cyto.), soluble nuclear (Nucl. sol.), and nuclear pellet (Nucl. pel.) fractions. Numbers represent quantified pixel intensity for HEXIM1 bands relative to un-transfected cells for the same fraction. Tubulin, SP1, and histone H4 were used as markers for each fraction and show the quality of the separations. HEXIM1 bands were first normalized to the pixel intensity of appropriate fraction controls (i.e.

*Figure 5 continued on next page*

*Figure 5 continued*

Cyto. normalized to tubulin, Nucl. sol. normalized to SP1, Nucl. pel. normalized to histone H4) prior to comparison between treatments. Two exposures of HEXIM1 from the same blot are shown. (B) Subcellular fractionation of HEK 293 T cells transfected with HIS-Ubiquitin. UBE2O-F was co-transfected as indicated and denaturing nickel pull-downs were used to detect HEXIM1 ubiquitination by anti-HEXIM1 western in the various subcellular fractions. Ubiquitinated species are indicated. (C) Subcellular localization of 0.25 µg of transfected HEXIM1-S or HEXIM1 ΔK-S plasmid in HEK 293 T cells. Numbers represent quantified pixel intensity for HEXIM1 (STREP) bands relative to wild-type expression for the same fraction. Bands were normalized as in (A).

DOI: https://doi.org/10.7554/eLife.31879.013

## Tat releases P-TEFb from cytoplasmic 7SK snRNP complexes for nuclear import and chromatin enrichment

Given the observation of HEXIM1 in the cytoplasm (*Figure 5A,B*) and the fact that UBE2O-dependent ubiquitination regulates HEXIM1's interactions with the P-TEFb–7SK snRNP complex (*Figure 6*), we wished to determine if the full 7SK snRNP was contained in the cytoplasmic compartment. Indeed, a native IP of endogenous HEXIM1 from cytoplasmic and soluble nuclear fractions revealed robust RNA-dependent interactions between HEXIM1 and proteins of the P-TEFb-7SK snRNP complex in the cytoplasm (*Figure 7A*), agreeing with a previous finding that a significant fraction of a 'large-form' of P-TEFb is in the cytoplasm (*Ramanathan et al., 1999*). It should be noted that fractions were generated under salt concentrations that do not disassemble the 7SK snRNP (*He et al., 2008*). Immunofluorescence confirmed that a significant fraction of the CCNT1 subunit of P-TEFb was localized to the cytoplasm (*Figure 7B*), consistent with a previous observation that CCNT1 and CDK9 are detected in the cytoplasm in unstimulated resting memory CD4[+] T cells, the target immune cells of HIV-1 infection (*Mbonye et al., 2013*). Further, the interaction of HEXIM1 with 7SK snRNA was also detected in the cytoplasm (*Figure 7C*, *Figure 7—source data 1*). Altogether, the results establish that, in addition to the known nuclear pool of 7SK snRNP, a cytoplasmic reservoir of the full 7SK complex exists.

We next examined whether Tat utilizes this cytoplasmic pool of inhibited P-TEFb. Since newly translated Tat is cytoplasmic, especially during acute infection (*Ranki et al., 1994*; *Kim et al., 2010*), and shuttles between the nucleus and cytoplasm (*Stauber and Pavlakis, 1998*), we reasoned Tat might alter the localization of P-TEFb. Indeed, Tat expressed in HeLa cells, which we observed in cytoplasmic and nuclear fractions (*Figure 7D*, STREP western), decreased the amount of CCNT1 and CDK9 in the cytoplasm four-fold with a concomitant increase in the nuclear, chromatin-containing pellet (*Figure 7D*). Notably, localization of HEXIM1, of which nearly 30% is in complex with P-TEFb in HeLa cells (*Barboric et al., 2007*; *Byers et al., 2005*), was unaltered, supporting that free, and not snRNP-bound, P-TEFb enters the nucleus. P-TEFb nuclear import can be observed at the lowest Tat expression levels tested, arguing against an overexpression phenotype.

With the Tat-dependent increase of P-TEFb in the chromatin fraction (*Figure 7D*) and the involvement of UBE2O in Tat-dependent P-TEFb release (*Figure 6*), we examined whether UBE2O was involved in P-TEFb chromatin recruitment, specifically at the HIV-1 promoter. As expected, Tat expression increased the enrichment of CDK9 at the viral promoter (*Figure 7E*, *Figure 7—source data 2*). Importantly, knockdown of UBE2O greatly inhibited Tat-dependent CDK9 recruitment to the integrated viral promoter, supporting a critical role of this ligase in regulating viral transcription. UBE2O knockdown also prevented CDK9 enrichment in the absence of Tat, supporting an endogenous role of UBE2O in regulating HEXIM1 ubiquitination and maintaining appropriate P-TEFb kinase levels on the HIV-1 promoter.

## The transcriptional inhibitor DRB re-localizes P-TEFb from the cytoplasm to the nucleus

Viruses often hijack existing host mechanisms; therefore, to investigate whether the nuclear import of P-TEFb freed from cytoplasmic 7SK snRNP might be a more general pathway to regulate transcriptional elongation, we used the CDK7/9 inhibitor 5,6-Dichloro-1-beta-D-ribofuranosylbenzimidazole (DRB), which blocks transcription elongation by decreasing RNAP II serine two phosphorylation (S2-P) (*Koga et al., 2015*). RNAP II S2-P levels dramatically decreased when HeLa cells were treated with DRB (*Figure 8A*). Remarkably, similar to Tat, DRB treatment induced extensive re-localization

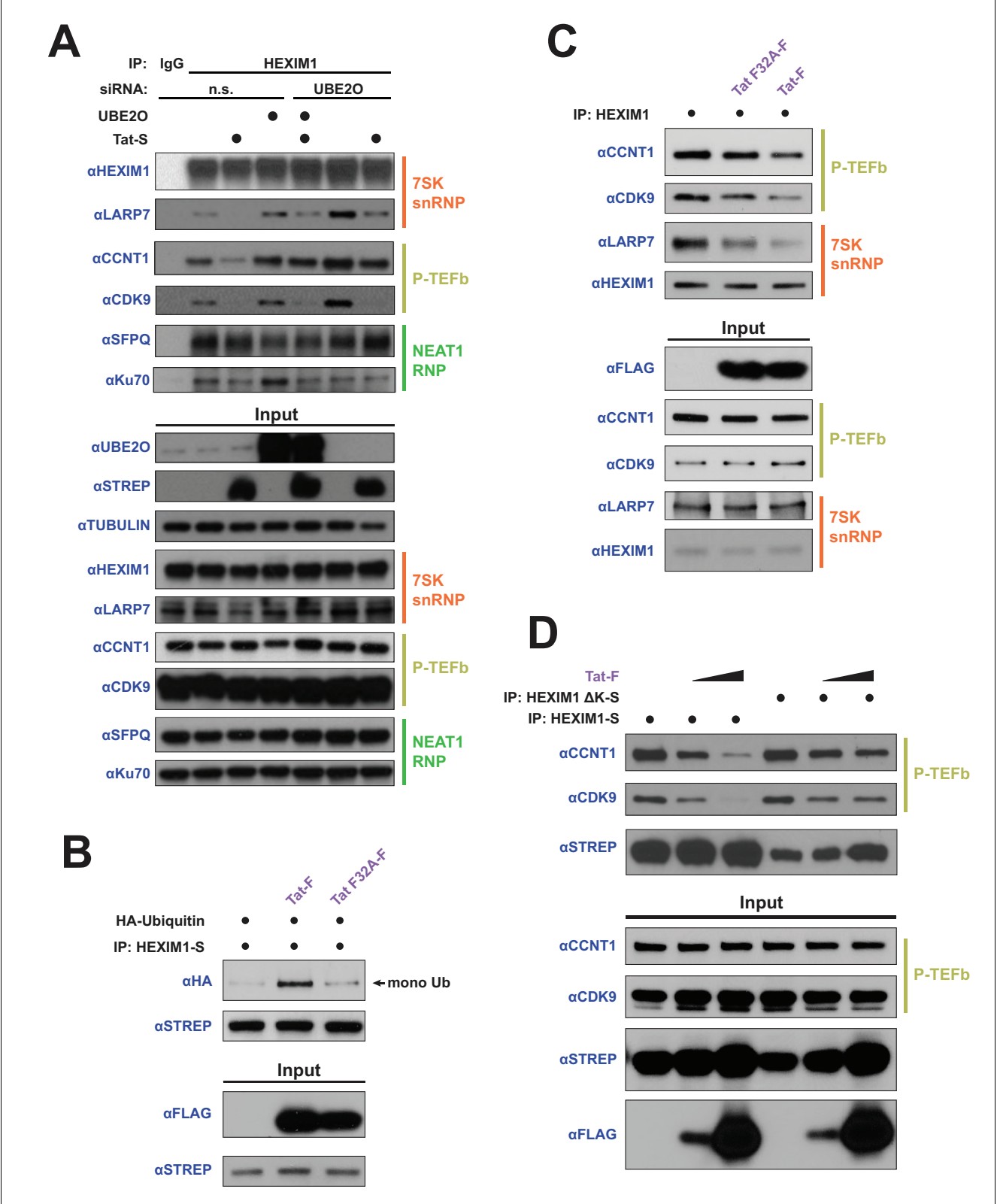

**Figure 6.** HEXIM1 ubiquitination is required for Tat-dependent release of P-TEFb from the inhibitory 7SK snRNP complex. (A) Endogenous IP of HEXIM1 in non-silencing (n.s.) and UBE2O RNAi cells, ±Tat expression. Rabbit IgG was used as a negative IP control. Western blots were performed with the specific endogenous antibodies indicated to detect HEXIM1 protein-protein interactions. (B) In vivo ubiquitination assay of HEXIM-S. Wild type Tat or the F32A mutant was co-transfected as indicated. HEXIM1 ubiquitination was monitored by anti-HA western. The monoubiquitinated HEXIM1

*Figure 6 continued on next page*

*Figure 6 continued*

species is indicated. (C) Endogenous IP of HEXIM1. Prior to lysis, cells were transfected with vector, wild type Tat, or Tat F32A. HEXIM1 interactions were detected by western blot with the indicated antibodies. (D) IP of HEXIM1-S wild type or ΔK mutant co-transfected with increasing Tat plasmid.

DOI: https://doi.org/10.7554/eLife.31879.014

of P-TEFb, but not HEXIM1, from the cytoplasm to the nuclear pellet, the same fraction containing reduced RNAP II S2-P. Supporting a role of UBE2O in this import pathway, DRB treatment also increased the association between UBE2O and endogenous HEXIM1 (*Figure 8B*). Given that DRB is a transcriptional inhibitor, one possibility is that P-TEFb nuclear import is part of a transcriptional stress response that senses and maintains proper polymerase phosphorylation in the nucleus. These results substantiate a model whereby re-localization of free P-TEFb to the nucleus is an important regulatory mechanism to sustain proper RNAP II S2-P levels.

## Discussion

Since HIV-1 has a small 9-kilobase genome encoding only 18 proteins, it has evolved ways to economically utilize its limited genetic information and rely heavily on host protein complexes to execute all stages of its life cycle. Here we show that the viral transcription factor Tat hijacks a new host partner, UBE2O, which ubiquitinates transcription machinery to enhance viral transcription. UBE2O binds Tat via an interaction surface distinct from the well-known surface used to bind P-TEFb (*Tahirov et al., 2010*). We discovered that a single lysine (Lys28) and adjacent phenylalanine (Phe32) in Tat are largely responsible for the interaction with UBE2O. The requirement of just a few critical residues for interaction helps further explain how Tat can rewire large host complexes for its function using relatively little genetic information. Previous structural work has implicated Lys28 and Phe32 in directly binding AFF4 of the SEC (*Schulze-Gahmen et al., 2014*) and thus may be an interaction 'hot spot'. Interestingly, while AFF4 is localized exclusively to the nucleus (*Figure 7D*), Tat recruits UBE2O to the 7SK snRNP in the cytoplasm and thus the 'hot spot' can be used to form complexes with different host partners in distinct cellular compartments.

A classic example of transcriptional regulation being coupled to signaling events in the cytoplasm is illustrated by the NF-κB pathway, where the application of a stimulus (e.g. TNFα) leads to the cytoplasmic degradation of inhibitor of κB (IκB) and subsequent nuclear import of NF-κB to activate gene expression (*Napetschnig and Wu, 2013*). By analogy, we find a significant pool of the inhibitory 7SK snRNP-P-TEFb complex in the cytoplasm, presumably reflecting the 'large' P-TEFb complexes previously described (*Ramanathan et al., 1999*), that serves as an inactive reservoir of P-TEFb. Surprisingly, Tat expression resulted in a significant redistribution of the P-TEFb subunits CCNT1 and CDK9, but not HEXIM1, from the cytoplasm to the histone-containing nuclear fraction, confirming that Tat can utilize this pool of inhibited P-TEFb. High FRET signal between Tat and CCNT1 in the cytoplasm (*Marcello et al., 2001*) and an intrinsic nucleocytoplasmic shuttling activity of CDK9 (*Napolitano et al., 2002*) support this hypothesis. Similar to NF-κB signaling, the P-TEFb nuclear import pathway also employs ubiquitination in that UBE2O-dependent modification of HEXIM1 is important for initial P-TEFb release from the 7SK snRNP (*Figure 6*). Intriguingly, UBE2O inhibits NF-κB activation by preventing the polyubiquitination of TRAF6 (*Zhang et al., 2013b*), suggesting that UBE2O is potentially intercalated in a signaling pathway that regulates stimulus-dependent NF-κB and P-TEFb nuclear import. An exciting future direction is to detail the additional cellular machinery that controls this multi-compartment signaling pathway.

We also demonstrated that another host protein, ZFP91, is important for Tat-dependent transcription (*Figure 1A*). ZFP91 is a classical C2H2 zinc finger (ZnF) protein, and while ZnF domains generally function as DNA-binding motifs, a previous report demonstrated that one ZFP91 ZnF domain can function as an atypical E3 ubiquitin ligase (*Jin et al., 2010*). We were unable to identify any substrate proteins for ZFP91 ligase activity in the Tat-P-TEFb transcriptional network. It will be interesting to determine whether ZFP91 mediates Tat co-activation through ubiquitination, and, if so, to identify the substrates of this activity. The previously identified target of ZFP91 ubiquitination was NF-κB-inducing kinase (NIK) (*Jin et al., 2010*), which is involved in activating the noncanonical NF-κB pathway. Therefore, in addition to the NF-κB binding sites in the HIV-1 promoter, Tat binds

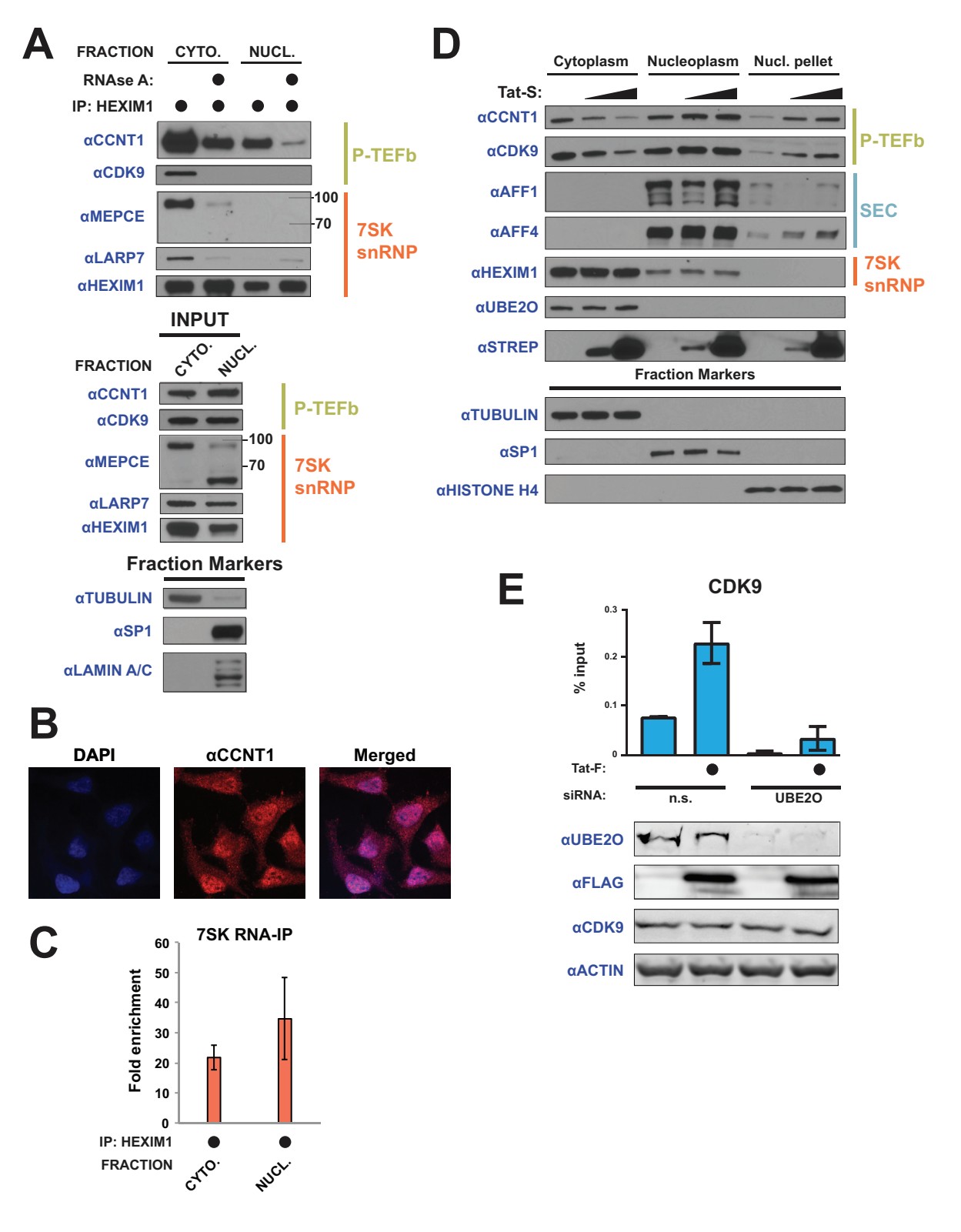

**Figure 7.** Tat utilizes UBE2O for the chromatin recruitment of P-TEFb released from a cytoplasmic pool of 7SK snRNP. (**A**) Endogenous HEXIM1 IP using cytoplasmic and soluble nuclear fractions. One sample from each fraction was treated with RNAse A prior to the IP to evaluate RNA-dependence of interactions. (**B**) Immunofluorescence analysis of endogenous CCNT1 in HeLa[provirusΔtat] cells, demonstrating nuclear and cytoplasmic staining. Nuclei were stained with DAPI. (**C**) RNA-IP assay of the 7SK snRNA. An endogenous HEXIM1 IP was performed using cytoplasmic and soluble nuclear

*Figure 7 continued on next page*

*Figure 7 continued*

fractions. RNA was then purified from the IPs and 7SK snRNA was detected by RT-qPCR. Fold enrichments were calculated as relative amounts of 7SK snRNA [HEXIM1 IP/rabbit IgG IP], with the average fold enrichments shown ± SEM for biological triplicates (purifications performed on different days from independently seeded cells). (D) HeLa cells were transfected with increasing amounts of Tat-S plasmid (0, 0.5, or 5 μg) followed by subcellular fractionation, and western blots of endogenous proteins were performed using the indicated antibodies. Fraction markers are shown to demonstrate the quality of the fractions. (E) Chromatin immunoprecipitation (ChIP) of CDK9 at the HIV-1 promoter in the HeLa[provirusΔtat] cell line, ±UBE2O RNAi, ±Tat expression. Values represent mean % input (IP/input) minus a non-specific IgG control from biological duplicate experiments (defined as independent transfections on different days). Error bars represent standard error. Western blots below demonstrate efficient UBE2O knockdown and Tat expression in the cells used for ChIP.

DOI: https://doi.org/10.7554/eLife.31879.015

The following source data is available for figure 7:

**Source data 1.** Source data for RIP assay of *Figure 7C*.
DOI: https://doi.org/10.7554/eLife.31879.016
**Source data 2.** Source data for ChIP experiment in *Figure 7E*.
DOI: https://doi.org/10.7554/eLife.31879.017

two NF-κB regulatory proteins (ZFP91 and UBE2O), demonstrating that HIV-1 has heavily co-opted this host transcription pathway to control its own gene expression.

Many diverse stimuli are known to release P-TEFb from the 7SK snRNP (*Amente et al., 2009*; *Bartholomeeusen et al., 2012*; *Biglione et al., 2007*; *Contreras et al., 2007*; *Nguyen et al., 2001*), so nuclear import of the free kinase as described here is likely a general mechanism to control

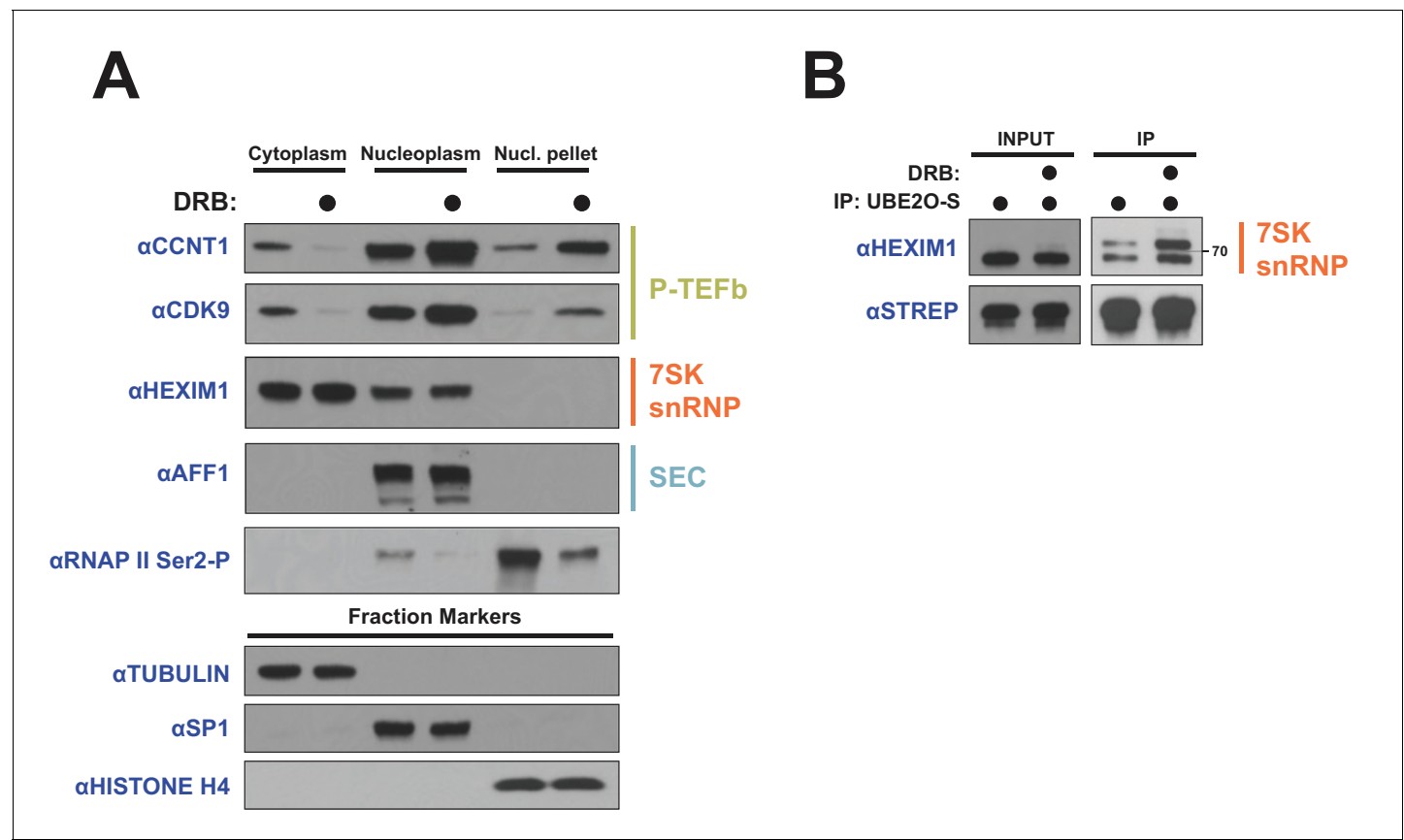

**Figure 8.** DRB releases P-TEFb from cytoplasmic 7SK snRNP. (A) HeLa cells were treated with 100 μM DRB or an equal volume of DMSO for 1 hr prior to subcellular fractionation and Western blotting. (B) HEK 293 T cells were transfected with UBE2O-S. Cells were then treated with 100 μM DRB or DMSO 1 hr prior to lysis and STREP IP. The endogenous HEXIM1 interaction was detected by anti-HEXIM1 western.
DOI: https://doi.org/10.7554/eLife.31879.018

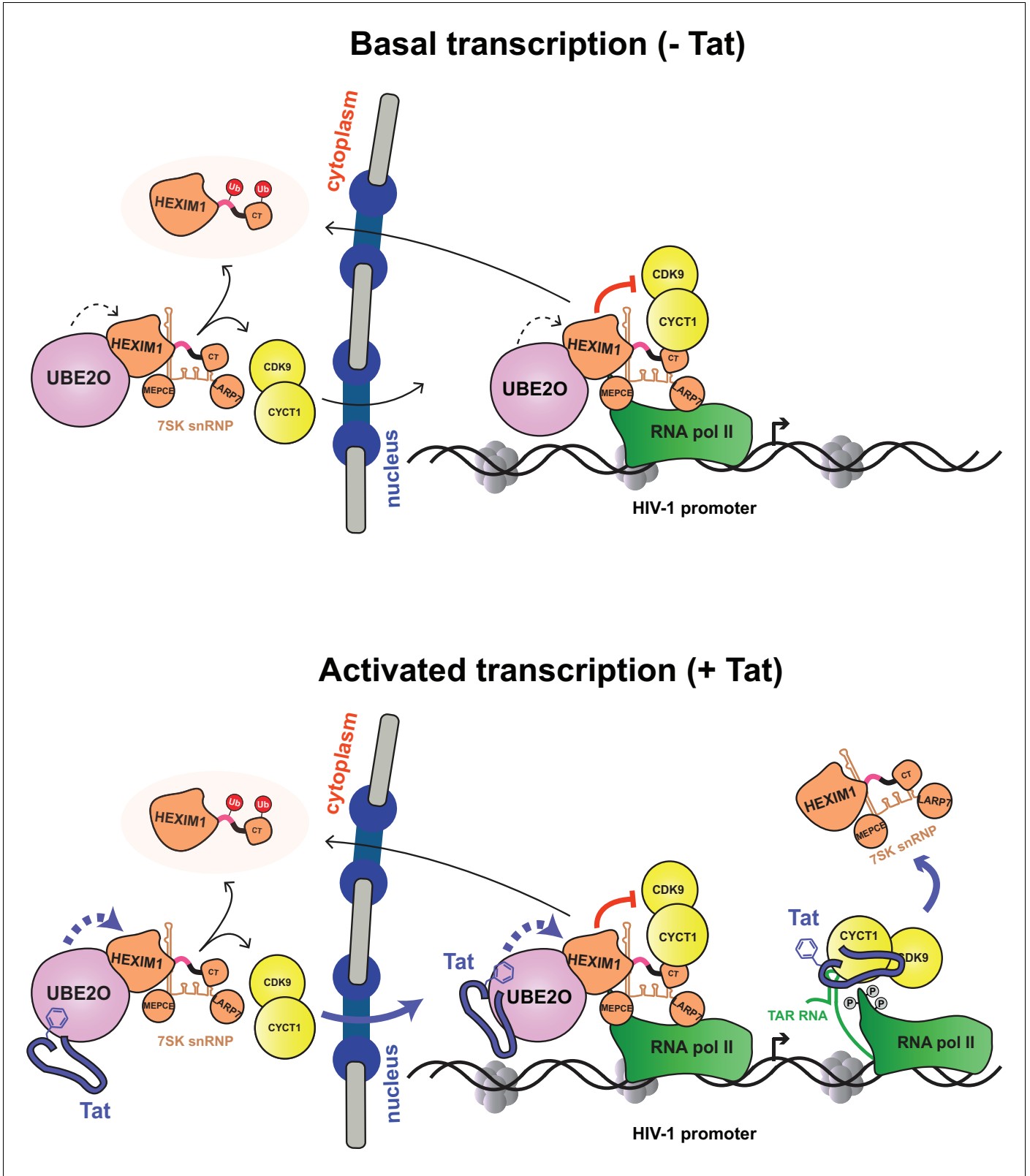

**Figure 9.** Model of basal and Tat-activated HIV-1 transcription. (Top) Basal transcription. The inhibited P-TEFb-7SK snRNP complex resides in the cytoplasm and bound to RNAP II at the viral promoter. Inhibition of P-TEFb kinase activity pauses the RNAP II-7SK snRNP complex at the promoter. UBE2O, also residing in the nucleus and cytoplasm, ubiquitinates HEXIM1, resulting in the sequestration of HEXIM1 in the cytoplasm and the nuclear import/retention of P-TEFb. This freed P-TEFb is then able to phosphorylate RNAP II, leading to low-level activation. (Bottom) Activated transcription.

*Figure 9 continued on next page*

Figure 9 continued

Once Tat is expressed, it can assemble with the inhibited P-TEFb complex at the HIV-1 promoter. The synthesis of TAR RNA results in the displacement of the inhibitory snRNP to activate the kinase activity of CDK9 (*D'Orso and Frankel, 2010*). Further, Tat enhances the association of UBE2O with HEXIM1, leading to increased HEXIM1 ubiquitination and P-TEFb release and nuclear import. Residue F32 in Tat is critically important for the UBE2O-dependent activation mechanism. Tat combines multiple P-TEFb targeting mechanisms to supply activated kinase to RNAP II at the HIV-1 promoter.

DOI: https://doi.org/10.7554/eLife.31879.019

transcription elongation and not restricted to this viral case. Indeed, treatment of cells with the CDK inhibitor DRB potently relocalized P-TEFb to the nuclear pellet fraction, phenocopying the Tat effect (*Figure 8A*). As many compounds releasing P-TEFb from 7SK snRNP are inhibitors, particularly of transcription, nuclear import of the freed kinase might be part of a transcriptional stress response pathway. In this model, a likely signal of transcriptional stress is the phosphorylation state of RNAP II. Any decrease in RNAP II phosphorylation is then sensed and relayed to the cytoplasm, where P-TEFb is freed from the inhibitory snRNP. It will be interesting to define which cellular machinery sense the phosphorylation status of RNAP II, and how this signal is relayed to the cytoplasm to ultimately restore RNAP II phosphorylation.

The hijacking of the UBE2O E3 ligase by Tat results in ubiquitination of the HEXIM1 protein of the 7SK snRNP and adds a new dimension to viral transcriptional regulation. Monoubiquitination by UBE2O does not cause protein degradation but rather stimulates two signaling outcomes, adding to the growing list of functions for ubiquitination (*Gatti et al., 2015*; *Thorslund et al., 2015*; *Werner et al., 2015*; *Yuan et al., 2014*). First, ubiquitinated HEXIM1 displays decreased enrichment in the chromatin fraction and accumulates in the cytoplasm (*Figure 5A,C*), which is analogous to a previously reported function of UBE2O ligase activity in regulating the localization of chromatin-associated proteins by increasing their cytoplasmic retention (*Mashtalir et al., 2014*). Second, HEXIM1 ubiquitination is important for the Tat-dependent release of P-TEFb from the 7SK snRNP and recruitment to the viral promoter for transcription activation. These two signaling outcomes of UBE2O activity are coupled in that ubiquitination both aids P-TEFb nuclear import and sequesters HEXIM1 in the cytoplasm, which could sustain P-TEFb kinase activity in the nucleus by limiting re-association of the free kinase with its inhibitor. Our findings reveal that the coordination of multiple localization pathways is involved in regulating transcription elongation.

The data presented here lead to an updated model of Tat activation (*Figure 9*). As shown previously (*D'Orso and Frankel, 2010*), the P-TEFb-7SK snRNP complex is recruited to the HIV-1 promoter in its basal state (without Tat), resulting in P-TEFb inhibition and accumulation of paused RNA polymerase II at the promoter, consequently poising the system for activation. In addition, UBE2O ubiquitinates nuclear and cytoplasmic HEXIM1, sequestering HEXIM1 in the cytoplasm and releasing some P-TEFb from the 7SK snRNP for nuclear import and retention. This activity of UBE2O supports basal, low-level HIV-1 gene expression.

Once Tat is expressed (*Figure 9*, bottom), it is also recruited to the HIV-1 promoter. The synthesis of TAR triggers Tat-dependent release of the 7SK snRNP, activating the P-TEFb kinase. Tat also binds UBE2O, further increasing HEXIM1 ubiquitination, thereby inducing P-TEFb release from the 7SK snRNP and promoting sequestration of HEXIM1 in the cytoplasm. Tat appears to target both the nuclear and cytoplasmic pools of the 7SK snRNP complex. P-TEFb that is released from the cytoplasmic pool enters the nucleus to sustain high-level transcriptional activation in Tat-expressing cells. The ability of Tat to hijack multiple reservoirs of P-TEFb helps explain its potent transcriptional activity. At the moment, it is unclear how Tat times the usage of the distinct fractions of P-TEFb–7SK snRNP. However, work presented here demonstrates that as Tat expression levels increase, more of the cytoplasmic P-TEFb reservoir is utilized, as evidenced by the dose response of Tat protein levels and the increase in P-TEFb nuclear import (*Figure 7D*). Given that DRB, a compound that induces stress and thus 7SK snRNP disassembly, also increases P-TEFb nuclear import and retention, it may be likely that the nuclear import of free P-TEFb kinase is a more general regulatory mechanism that is not restricted to HIV-1 transcription.

# Materials and methods

## Key resources table

| Reagent type (species) or resource | Designation | Source or reference | Identifiers | Additional information |
|---|---|---|---|---|
| Recombinant DNA reagent | UBE2O-FLAG (plasmid) | this paper | | vector:pcDNA4/TO, insert: full-length Homo sapiens UBE2O |
| Recombinant DNA reagent | HEXIM1-STREP (plasmid) | this paper | | vector:pcDNA4/TO, insert: full-length Homo sapiens HEXIM1 |
| Sequence-based reagent | ZFP91 siRNA #1 | Qiagen | SI05109223 | |
| Sequence-based reagent | ZFP91 siRNA #2 | Qiagen | SI05109230 | |
| Sequence-based reagent | ZFP91 siRNA #3 | Qiagen | SI05150985 | |
| Sequence-based reagent | UBE2O siRNA #1 | Qiagen | SI04153863 | |
| Sequence-based reagent | UBE2O siRNA #2 | Dharmacon | J-008979–05 | |
| Sequence-based reagent | UBE2O siRNA #3 | Dharmacon | J-008979–06 | |
| Sequence-based reagent | CDK9 siRNA | Qiagen | SI00605066 | |
| Sequence-based reagent | CCNT1 siRNA | Qiagen | SI02625707 | |
| Sequence-based reagent | Non-silencing siRNA | Qiagen | SI03650318 | |
| Cell line (Homo sapiens) | HeLa (provirusΔtat) | PMID: 28345603 | | |
| Peptide, recombinant protein | HA-ubiquitin | this paper | | purified from E. coli BL21 |
| Peptide, recombinant protein | HA-ΔK ubiquitin | this paper | | purified from E. coli BL21 |
| Peptide, recombinant protein | UBE2O | this paper | | purified from HEK 293 T cells with high salt |
| Peptide, recombinant protein | UBE2O, catalytically inactive | this paper | | purified from HEK 293 T cells with high salt |
| Peptide, recombinant protein | HEXIM1 | this paper | | purified from HEK 293 T cells with high salt |
| Commercial assay or kit | HiPerFect transfection reagent | Qiagen | | used for transfecting siRNA |
| Commercial assay or kit | PolyJet Transfection reagent | Signa Gen | | used for transfecting DNA |
| Commercial assay or kit | STREP-tactin superflow resin | IBA Lifesciences | | |
| Antibody | anti-HA | Covance | MMS-101P RRID:AB_2314672 | 1:2000 dilution |
| Antibody | anti-HEXIM1 | Abcam | ab25388 RRID:AB_2233058 | 1:2000 dilution, also used for IP |
| Antibody | anti-CDK9 | Santa Cruz Biotechnology | sc-13130 RRID:AB_627245 | used for ChIP |
| Antibody | anti-IgG | Millipore | 12–371 RRID:AB_145840 | used for ChIP |
| Antibody | anti-IgG | Santa Cruz Biotechnology | sc-2027 RRID:AB_737197 | For IP |

*Continued on next page*

*Continued*

| Reagent type (species) or resource | Designation | Source or reference | Identifiers | Additional information |
|---|---|---|---|---|
| Antibody | anti- RNA polymerase II | Santa Cruz Biotechnology | sc-899x RRID:AB_632359 | 1:2000 |
| Antibody | anti- CCNT1 | Santa Cruz Biotechnology | sc-8127 RRID:AB_2073892 | 1:2000 |
| Antibody | anti- CCNT1 | Santa Cruz Biotechnology | sc-10750 RRID:AB_2073888 | 1:2000 |
| Antibody | anti- MEPCE | Santa Cruz Biotechnology | sc-82542 RRID:AB_2266399 | 1:1000 |
| Antibody | anti- SP1 | Santa Cruz Biotechnology | sc-14027 RRID:AB_2171049 | 1:5000 |
| Antibody | anti-AFF1 | Bethyl | A302-344A RRID:AB_1850255 | 1:1000 |
| Antibody | anti- AFF4 | Bethyl | A302-538A RRID:AB_1998985 | 1:1000 |
| Antibody | anti- UBE2O | Bethyl | A301-873A RRID:AB_1309799 | 1:1000 |
| Antibody | anti-Ku70 | Bethyl | A302-624A-T RRID:AB_10554672 | 1:1000 |
| Antibody | anti- SFPQ | Bethyl | A301-322A-T RRID:AB_937995 | 1:1000 |
| Antibody | anti-LARP7 | Aviva | ARP40847_P050 RRID:AB_1294403 | 1:2000 |
| Antibody | anti-LARP7 | Santa Cruz Biotechnology | sc-515209 RRID:AB_2728652 | 1:2000 |
| Antibody | anti-HEXIM1 | Everest | EB06964 RRID:AB_2118260 | 1:5000 |
| Antibody | anti-CDK9 | Santa Cruz Biotechnology | sc-8338 RRID:AB_2260303 | 1:2000 |
| Antibody | anti-CDK9 | ProMab | 30212 RRID:AB_2728653 | 1:2000 |
| Antibody | anti-TUBULIN | Sigma-Aldrich | T6199 RRID:AB_477583 | 1:10,000 |
| Antibody | anti-FLAG | Sigma-Aldrich | F1804 RRID:AB_262044 | 1:5000 |
| Antibody | anti-RNA pol II phosphor-Ser2 | Abcam | ab5095 RRID:AB_304749 | 1:2000 |
| Antibody | anti-STREP-HRP | Millipore | 71591 RRID:AB_10806716 | 1:5000 |
| Antibody | anti-histone H4 | Active Motif | 39269 RRID:AB_2636967 | 1:5000 |

## Cell culture, plasmids, and reagents

HeLa$^{provirus\Delta tat}$, HeLa (RRID:CVCL_0030), and HEK 293T (RRID:CVCL_0063) cells were maintained in DMEM (10% FBS, 1% pen/strep) at 37C with 5% $CO_2$ and discarded after 25–30 passages. HeLa and HEK 293 T cells were purchased from ATCC as authenticated cell lines. The HeLa$^{provirus\Delta tat}$ cell line was generated from an early frozen stock of authenticated HeLa cells and was validated by Southern blot, which has been detailed previously (*Faust et al., 2017*). Symptoms for mycoplasma contamination were not observed, therefore no test for mycoplasma contamination was performed. ZFP91 and UBE2O were cloned into pcDNA4-TO (Thermo) with a C-terminal 3x-FLAG tag. ZFP91, UBE2O, CCNT1, CDK9, HEXIM1, LARP7, MEPCE, and HIV-1 HXB2 Tat were cloned into pcDNA4-TO with a C-terminal 2xSTREP tag. Human ubiquitin was cloned into pcDNA4-TO containing an N-terminal HA tag. All HEXIM1 and HIV-1 Tat point mutants were generated by standard quick-change mutagenesis. The HEXIM1 ΔK mutants were ordered as a gBlock gene fragment from IDT with restriction sites

and cloned into appropriate vectors. DNA was transfected with Poly Jet (Signa Gen) and siRNA was transfected with HiPerFect (Qiagen) or RNAi MAX (Thermo). All transfections were performed according to manufacturer's protocol. MG132 was purchased from Selleckchem (S2619). DRB was purchased from Abcam (ab120939). 2x and 4x Laemmli buffer were purchased from BioRad. Human E1 protein was a kind gift from the laboratory of John Gross.

## Antibodies

Normal rabbit IgG (sc-2027), RNA polymerase II (sc-899x), CCNT1 (sc-8127 and sc-10750), MEPCE (sc-82542), and SP1 (sc-14027) were purchased from Santa Cruz Biotechnology. AFF1 (A302-344A), AFF4 (A302-538A), UBE2O (A301-873A), Ku70 (A302-624A-T), and SFPQ (A301-322A-T) antibodies were purchased from Bethyl. The HA antibody (MMS-101P) was purchased from Covance. LARP7 antibodies were purchased from Aviva (ARP40847_P050) and Santa Cruz (sc-515209). HEXIM1 antibodies were purchased from Abcam (ab25388) and Everest (EB06964). CDK9 antibodies were purchased from Santa Cruz (sc-8338 and sc13130) and ProMab (30212). The tubulin (T6199) and FLAG (F1804) antibodies were purchased from Sigma. The RNA pol II phosphor-Ser2 (ab5095) was purchased from Abcam. The STREP-HRP antibody (71591) and IgG (12-371) were purchased from Millipore. The histone H4 antibody (39269) was purchased from Active Motif.

## RNAi knockdown of Tat host factors

HeLa$^{provirus\Delta tat}$ cells were transfected with non-silencing or specific siRNAs (Qiagen) at 5–10 nM final concentrations using HiPerFect. After 48 hours, cells were transfected with 0.5 ng Tat-expressing plasmid and 1 ng CMV-firefly luciferase plasmid as a transfection control using PEI. After an additional 48 hr, cells were lysed in cell lysis buffer (5 mM Tris-HCl pH 8.0, 85 mM KCl, 0.5% IGEPAL CA-630). Cell-associated HIV-1 p24 production was detected by p24 ELISA. In parallel, 10 μL of lysate was used in a standard luciferase assay. All p24 values were normalized to firefly luciferase activity for each siRNA to generate a relative Tat activity score, defined as [p24$^{host\ RNAi}$/FFL$^{host\ RNAi}$] / [p24$^{N.S.\ RNAi}$/FFL$^{N.S.\ RNAi}$]. For analysis of protein knockdown, 2x Laemmli sample buffer was added to siRNA-treated cells and then loaded onto gels. Knockdowns were determined by western blot with the indicated antibodies. siRNAs:

ZFP91 #1: CTGCGGCACACTTATCTTCAA (Qiagen, SI05109223)
ZFP91 #2: CAGCTCTTAAAGTGAGGGTTA (Qiagen, SI05109230)
ZFP91 #3: CCGCGACTCCTATGCATAGAA (Qiagen, SI05150985)
UBE2O #1: TCCCGGGACCATTCCATGGAA (Qiagen, SI04153863)
UBE2O #2: GACUUUAGGUUCCGUACAA (Dharmacon, J-008979–05)
UBE2O #3: GUUGUAGAGUUGAAAGUUA (Dharmacon, J-008979–06)
CDK9: TAGGGACATGAAGGCTGCTAA (Qiagen, SI00605066)
CCNT1: AGGCTTTGAACTAACAATTGA (Qiagen, SI02625707)
Non-silencing (N.S.): Qiagen All-Star negative control (SI03650318)

## Denaturing in vivo ubiquitination assay

HEK 293T or HeLa cells were transfected using PolyJet with 7 μg total DNA containing STREP-tagged bait protein and HA-Ubiquitin for denaturing STREP IPs or HIS-Ubiquitin for denaturing HIS IPs. After 30 hr, cells were lysed in an equal volume SDS Lysis Buffer (2% SDS, 50 mM Tris-HCl pH 8.0, 150 mM NaCl, 0.5 mM DTT) for denaturing STREP IPS or PBS + 6M guanidinium-HCl for denaturing HIS IPs and boiled for 10 min. Boiled lysates were sonicated on a Fisher Sonic Dismembrator Model 500 and centrifuged in a microfuge at full speed. Cleared lysates were incubated with Strep-tactin Superflow Agarose or HisPur nickel resin for 3 hr at room temperature. For STREP elution, the resin was washed with RIPA buffer and bound material was eluted with 1x STREP elution buffer (IBA-GmbH). For HIS elution, resin was washed with PBS + 0.2% Triton X-100 and bound material was eluted with PBS + 250 mM imidazole. Ubiquitination of immunoprecipitated material was detected by HA western blot for STREP IP or with endogenous antibodies for HIS IP. For RNAi-ubiquitination experiments, cells were first transfected with siRNA for 48 hr prior to DNA transfection. For MG132 treatment, cells that had been transfected with DNA for 24 hr were then incubated with 2.5 μM MG132 for an additional 16 hr before harvesting.

## Cloning, expression, and purification of recombinant proteins

Human wild type and ΔK ubiquitin with an N-terminal HA tag were cloned into the pGEX6p-1 vector (GE Healthcare) and expressed as a fusion protein with an N-terminal GST tag. The fusion proteins were expressed in *E. coli* BL21 (DE3). The GST-fusion proteins were digested on column with PreScission protease. The released ubiquitin proteins were collected and further purified by Superdex 200 Increase (GE Healthcare).

## In vitro ubiquitination assay

Assays were performed in a total volume of 30 μL with 0.1 μM recombinant human E1, 0.1 μM wild type UBE2O or catalytically inactive UBE2O (both wild-type UBE2O and UBE2O CD were purified from HEK 293 T cells with high salt), 5 μM HA-tagged ubiquitin (wild type or ΔK) and 0.2 μM HEXIM1 (purified from HEK 293 T cells) in reaction buffer containing 50 mM Tris-HCl pH7.4, 100 mM NaCl, 1 mM DTT, 5 mM $MgCl_2$, 4 mM ATP at 37°C with constant shaking for 2 hr. Samples were quenched with 4x Laemmli sample buffer (containing 2-Mercaptoethanol) before analysis by SDS-PAGE. Ubiquitination of HEXIM1 was verified by immunoblotting with anti-HEXIM1 antibody.

## RNA elongation assay

HeLa$^{provirusΔtat}$ cells were transfected with siRNAs to obtain 10 nM final concentration. After 48 hr of RNAi, cells were transfected with a Tat-expressing plasmid for an additional 10 hr. RNA was purified on Zymo Quick-RNA Mini Prep columns (R1054) according to the manufacturer's protocol, DNAse-treated with TURBO DNAse (Thermo), and used in RT-PCR reactions with a primer immediately downstream of the HIV-1 TAR sequence to detect short RNA species and oligo dT to detect Tat-activated, poly-adenylated messages. RNA abundance was then quantified by qPCR, and HIV-1 RNA levels (short and long) were normalized to RPLP0 RNA levels (housekeeping gene). Values are reported as relative Tat activity, defined as [HIV-1 RNA$^{host\ RNAi}$/RPLP0 RNA$^{host\ RNAi}$] / [HIV-1 RNA$^{N.S.\ RNAi}$/RPLP0 $^{N.S.\ RNAi}$]. Primers used in qPCR reactions are:

HIV-1 Proximal F 5'-CTCTCTGGTTAGACCAGATCTGA-3'
HIV-1 Proximal R 5'-GTGGGTTCCCTAGTTAGCCAGA-3'
HIV-1 Distal F 5'-CACATGAAGCAGCACGACTT-3'
HIV-1 Distal R 5'-GGTCTTGTAGTTGCCGTCGT-3'
RPLP0 F 5'-GCTGCTGCCCGTGCTGGTG-3'
RPLP0 R 5'-TGGTGCCCCTGGAGATTTTAGTGG-3'

## Affinity purification and western blot

HEK 293T or HeLa cells were transfected with 7 μg total DNA containing STREP-tagged or FLAG-tagged bait protein. After 40 hr, cells were collected, lysed in Lysis Buffer (50 mM Tris-HCl pH 8.0, 150 mM NaCl, 2 mM $MgCl_2$, 0.5% Triton X-100, 0.2% Na-deoxycholate, Roche protease inhibitor cocktail) and rocked for 30 min at 4°C. Lysate was cleared and incubated with Streptactin Superflow resin slurry (IBA-GmBH) or magnetic M2-FLAG resin slurry (Sigma) at 4°C. The resin was washed with WASH buffer (50 mM Tris-HCl pH 8.0, 150 mM NaCl, 2 mM $MgCl_2$, 0.5% Triton X-100) and eluted with 1x STREP elution buffer (IBA-GmBH) diluted in WASH buffer or FLAG elution buffer (200 μg/mL 3x FLAG peptide, diluted in WASH buffer), shaking at RT for 30 min. The eluate was mixed 1:1 with 2x Laemmli sample buffer, boiled, and loaded on a SDS-PAGE gel. Proteins were detected with specific antibodies.

### RNA-IP

A fraction of the elution from a STREP IP was resuspended in RNA Lysis Buffer (Zymo). RNA was then processed with Zymo Quick-RNA Mini Prep kit as detailed above. Primers specific for the 7SK snRNA were used in RT-qPCR reactions (7SK F: 5'-GGATGTGAGGCGATCTGGC-3', 7SK R: 5'-AAAAGAAAGGCAGACTGCCAC-3')

### High-salt IP

For high-salt purification of UBE2O WT and CD from HEK 293 T cells, the same lysis and IP was performed as above. For washing, beads were washed 4x in High Salt Wash Buffer (25 mM Tris-HCl pH 8.0, 0.1% Triton X-100, 2 mM $MgCl_2$, 1 M NaCl, 0.5 mM DTT) and 2x in IP Dilution Buffer (25 mM

Tris-HCl pH 8.0, 0.1% Triton X-100, 2 mM MgCl$_2$, 150 mM NaCl, 0.5 mM DTT) and eluted as described.

## Tandem IP

2 × 15 cm plates of HEK 293 T cells were transfected per condition with either 5 µg of Tat-STREP and 5 µg UBE2O-FLAG or 5 µg of GFP-STREP-FLAG. The same lysis and IP was performed as above for the first STREP IP. The STREP elution was then incubated with FLAG-M2 resin overnight at 4C. FLAG resin was wash 4x with FLAG wash buffer (50 mM Tris pH 8.0, 2 mM MgCl$_2$, 0.1% Triton X-100, 150 mM NaCl). Bound material was eluted with the same FLAG elution buffer as above.

## Endogenous IP

HeLa or HEK 293 T cells were lysed in Lysis Buffer. Cleared cell lysates were incubated with control IgG or specific antibodies (1–10 µg) for 3 hr. Protein A Dynabeads slurry (Life Technologies) was washed and added to antibody-lysate mixture for an additional 3 hr. Dynabeads were then washed with WASH buffer. For elution, 50 µL of 2x Laemmli Buffer (BioRad) was added to beads and incubated at 65C for 15 min, with occasional mixing. After elution, 1.5 µL of BME was added to eluate, which was then processed by SDS-PAGE and Western blotting.

## Endogenous RNA-IP

Immunocomplexes were eluted with 50 µL of RNA elution buffer (10 mM EDTA pH 8.0, 1% SDS, 50 mM Tris-HCl pH 8.0) for 15 min at 65°C. RNA was purified from the elution on Quick-RNA MiniPrep columns as detailed above. 4 µL of DNAse-treated RNA was used in an RT reaction with primers specific for the 7SK snRNA and quantitated by qPCR.

## Affinity purification-mass spectrometry

AP-MS experiments were performed as previously described (*Jäger et al., 2011a*; *2011b*). Slight modifications were as follows: HEK 293 T cells were transfected with 5–20 µg plasmid complexed with PolyJet Reagent. After 40 hr, cells were harvested and washed in PBS, and then resuspended in Lysis Buffer. Following incubation at 4°C for 30 min on a tube rotator, lysates were centrifuged for 20 min at 4°C. Cleared lysates and 2 µg of 1x FLAG peptide (Sigma) were added to 20 µL anti-FLAG M2 magnetic beads and rotated for 2 hr at 4°C. FLAG beads were sequentially washed 3x with Wash Buffer (0.05% NP-40, 50 mM Tris-HCl pH 7.4, 150 mM NaCl, 1 mM EDTA) and 1x with Wash Buffer without NP-40. Beads were incubated with Elution Buffer (50 mM Tris-HCl, pH 7.4, 150 mM NaCl, 1 mM EDTA, 0.05% RapiGest SF Surfactant (Waters)) on a vortex mixer at room temperature for 30 min. Eluates (10 µL) were processed and analyzed by mass spectrometry as previously described (*Davis et al., 2015*; *Mirrashidi et al., 2015*).

## Gel filtration

HEK 293 T cells were lysed in Lysis Buffer. Cell extracts were filtered through a 0.22 µm filter and loaded on a Superdex 200 Increase 10/300 GL (GE Healthcare) in 1x phosphate-buffered saline containing 1 mM DTT. The flow rate was 0.7 mL/min and 0.5 mL fractions were collected after the elution of the void volume. Each fraction was subjected to western blot analysis.

## Identification of ubiquitinated HEXIM1 lysines by MS

HEK 293 T cells in 11 × 15 cm plates were transfected with 1 µg of HEXIM1-S, 5 µg of UBE2O-F, and 2 µg of HA-Ub (to 15 µg DNA total with pBluescript) by PolyJet. After 48 hr, a denaturing STREP IP was performed on the lysate. Sample eluate was digested with trypsin or Arg-C (according to manufacturer's protocol) and analyzed by mass spectrometry as previously described (*Kane et al., 2015*) with slight modifications. Briefly, the raw data was matched to protein sequences by the MaxQuant algorithm (version 1.5.2.8) (*Cox and Mann, 2008*). Data were searched against a database containing SwissProt Human (downloaded 12/2014), and the APEX2 sequence, concatenated to a decoy database where each sequence was randomized in order to estimate the false discovery rate (FDR). The data were filtered to obtain a peptide, protein, and site-level false discovery rate of 0.01.

## Subcellular fractionation

HEK 293T or HeLa cells were collected and the pelleted cell volume (PCV) was measured. For DRB treatment, cells were treated with DMSO control or 100 μM DRB for 1 hr prior to harvesting. Cells were washed in 3x PCV of Buffer A (10 mM Tris-HCl, pH 8.0, 1.5 mM $MgCl_2$, 10 mM KCl, 0.5 mM DTT, Roche protease inhibitor cocktail), resuspended in 5x PCV Buffer A, and allowed to swell on ice. Triton X-100 was then added to the cells to achieve a final concentration of 0.5% and rocked for 1 min. The cellular material was passed through a 25G needle and spun at 1500 x g for 5 min. The supernatant was collected as the cytoplasmic fraction. The nuclei were resuspended in ¼ pelleted nuclear volume (PNV) of Buffer B (20 mM Tris-HCl, pH 8.0, 25% glycerol, 1.5 mM $MgCl_2$, 20 mM KCl, 0.2 mM EDTA, 0.5 mM DTT, Roche protease inhibitors). ¼ PNV of Buffer C (Buffer B with 1.2 M KCl) was then added drop-wise to nuclei, which were then rocked for 30 min. The suspension was centrifuged and the supernatant was collected as the high salt nuclear extract (nucleoplasm). The nuclear pellet was then resuspended in RIPA buffer and sonicated. The sonicated lysate was spun at full speed and the supernatant was collected as the nuclear pellet. A Bradford assay (BioRad) was used to determine the protein concentration in each fraction. 10 μg of protein from each fraction was analyzed by SDS-PAGE and western blot.

## Immunofluorescence

HeLa$^{provirusΔtat}$ cells on coverslips were fixed in 4% formaldehyde for 20 min, quenched with 0.15 M glycine, and permeabilized with 0.1% Triton X-100 for 15 min. Permeabilized cells were blocked with blocking buffer (10 mM Tris-HCl pH 7.5, 150 mM NaCl, 2% BSA, 10% goat serum) for 1 hr, and then incubated with anti-CCNT1 antibody (sc-10750) at dilution of 1:100 for 1 hr. Cells were then washed with PBS and incubated with goat anti-rabbit IgG-Alexa Fluor 555 (1:1000) and 100 ng/μL DAPI in blocking buffer. Cover slips were washed in PBS and mounted on slides with ProLong Gold antifade mounting solution (Thermo). Confocal images were obtained with a spinning disk confocal microscope (Nikon Ti-E microscope equipped with Yokagawa CSU-22) by using the 561 and 405 nm excitation for Alexa Fluor 555 epifluorescence and 4,6-diamidino-2-phenylindole (DAPI), respectively.

## Chromatin immunoprecipitation (ChIP)-qPCR

HeLa$^{provirusΔtat}$ cells were transfected with either N.S. or UBE2O siRNA to obtain a 10 nM final concentration. After 48 hr, cells were transfected with a Tat-expressing plasmid (pcDNA4\TO-Tat:FLAG) for the +Tat condition and a mock expressing plasmid control (pcDNA4\TO:FLAG) for the –Tat condition for an additional 15 hr. $8.0 \times 10^6$ cells per ChIP were grown on 10 cm plates and were crosslinked with 0.5% formaldehyde for 10 min followed by addition of 150 mM glycine for 5 min to quench the crosslinking reaction. Cells were harvested and lysed for 30 min at 4°C with rotation in Farnham lysis buffer (5 mM PIPES pH 8.0, 85 mM KCl, 0.5% NP-40, 1 mM PMSF, 1x EDTA-free protease inhibitor tablet (Roche)) at a concentration of $10^7$ cells/mL. Nuclei were collected and resuspended in cold Szak's RIPA buffer (50 mM Tris-HCl pH 8.0, 1% NP-40, 150 mM NaCl, 0.5% Na-Deoxycholate, 0.1% SDS, 2.5 mM EDTA, 1 mM PMSF, 1x EDTA-free protease inhibitor tablet (Roche)) to a final concentration of $2.5 \times 10^7$ nuclei/mL. Nuclei were sonicated for 60 cycles (30 s ON, 30 s OFF) on a waterbath sonicator (Diagenode) to obtain DNA fragment sizes between 150 and 500 bp. Sheared chromatin was pre-cleared with Protein G beads for 1 hr at 4°C. Protein G beads (100 μL of 10 mg/mL stock, Thermo Fisher) were conjugated with 10 μg of the following antibodies: CDK9 (Santa Cruz Biotechnology, sc13130) and IgG (Millipore, 12–371). Antibody bound beads were blocked with 5% BSA +RIPA for 1 hr. Pre-cleared sheared chromatin ($8 \times 10^6$ nuclei) was added to the blocked antibody bound beads and incubated overnight. Washes, elutions, and de-crosslinking was performed as previously described (*McNamara et al., 2016*). Primers used for qPCR amplification of the HIV promoter (HIV (+141) Fwd: 5'-GCTTAAGCCTCAATAAAGCTTGCC TTGAG-3', HIV (+141) Rev: 5'-GTCCTGCGTCGAGAGATCTCCTCTG-3').

## Acknowledgements

We thank Dr. JJ Miranda, Dr. John Gross, and members of the Frankel laboratory for critically reading the manuscript and Dr. Orly Laufman for technical assistance with the immunofluorescence. Research reported in this publication was supported by the National Institute of Allergy and

Infectious Diseases of the NIH under award numbers R01AI114362 and Welch Foundation grant number I-1782 (to ID) and by the National Institute of General Medical Sciences of the NIH under number P50GM082250 (to NJK and ADF). CWB was supported by a pre-doctoral award from the National Science Foundation Graduate Research Fellowship under Grant No. 2016220513.

## Additional information

### Funding

| Funder | Grant reference number | Author |
| --- | --- | --- |
| National Institute of General Medical Sciences | P50GM082250 | Nevan J Krogan<br>Alan D Frankel |
| National Institute of Allergy and Infectious Diseases | R01AI114362 | Iván D'Orso |
| Welch Foundation | I-1782 | Iván D'Orso |

The funders had no role in study design, data collection and interpretation, or the decision to submit the work for publication.

### Author contributions

Tyler B Faust, Conceptualization, Investigation, Visualization, Writing—original draft, Writing—review and editing; Yang Li, Conceptualization, Investigation, Visualization, Writing—review and editing; Curtis W Bacon, Bhargavi Jayaraman, Investigation, Visualization, Writing—review and editing; Gwendolyn M Jang, Conceptualization, Investigation, Writing—review and editing; Amit Weiss, Billy W Newton, Investigation, Visualization; Nevan J Krogan, Supervision, Funding acquisition, Visualization, Writing—review and editing; Iván D'Orso, Visualization, Writing—review and editing; Alan D Frankel, Conceptualization, Resources, Supervision, Funding acquisition, Visualization, Writing—original draft, Project administration, Writing—review and editing

### Author ORCIDs

Yang Li (iD) https://orcid.org/0000-0003-0216-0350
Curtis W Bacon (iD) https://orcid.org/0000-0001-6454-5470
Iván D'Orso (iD) http://orcid.org/0000-0002-1409-2351
Alan D Frankel (iD) http://orcid.org/0000-0002-2525-9508

### Decision letter and Author response

Decision letter https://doi.org/10.7554/eLife.31879.026
Author response https://doi.org/10.7554/eLife.31879.027

## Additional files

### Supplementary files

• Supplementary file 1. Full HEXIM1, MEPCE, and LARP7 AP-MS lists. Each tab is a biological replicate FLAG affinity purification for the indicated protein.
DOI: https://doi.org/10.7554/eLife.31879.020

• Supplementary file 2. HEXIM1-Ub diGly data, related to *Figure 4*. Tabs show raw data for all detected unmodified and modified HEXIM1 peptides (HEXIM1 tab) or the sum of MS/MS counts for peptides generated from trypsin or Arg-C treatment (HEXIM1 summ. tab).
DOI: https://doi.org/10.7554/eLife.31879.021

• Transparent reporting form
DOI: https://doi.org/10.7554/eLife.31879.022

### Data availability

All data generated or analysed during this study are included in the manuscript and supporting files.

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
