## [Decision Letter]

Thank you for submitting your article "The HIV-1 Tat protein recruits a cytoplasmic ubiquitin ligase to reorganize the 7SK snRNP for transcriptional activation" for consideration by *eLife*. Your article has been evaluated by Kevin Struhl (Senior Editor) and two reviewers, one of whom is a member of our Board of Reviewing Editors. The following individual involved in review of your submission has agreed to reveal his identity: Andrew Rice (Reviewer #2).

The reviewers have discussed the reviews with one another and the Reviewing Editor has drafted this decision to help you prepare a revised submission.

Summary:

This manuscript by Faust and colleagues investigates the role of the Tat-associated ubiquitin ligase UBE20 in Tat/7SK RNP/P-TEFb function. Tat was found to direct UBE20 to ubiquitinate HEXIM1 in the 7SK RNP complex within the cytoplasm. This modification was found to increase HEXIM1 levels in the cytoplasm and to eject the LARP7 subunit from the 7SKRNP complex. Tat-UBE20 also appears to promote localization of P-TEFb (CDK9/CCNT1) to the nucleus, where this transcription factor activates elongation of HIV LTR-directed transcription.

Essential revisions:

The authors identify UBE2O as a cytoplasmic E3 ubiquitin ligase that associates with HIV-1 Tat, and report that it monoubiquitylates the HEXIM1 protein and affects its association with LARP7 and release of P-TEFb to the nucleus, to affect HIV-1 transcription. The data presented are credible but rely on over-expressed proteins and in vitro ubiquitination assays. In addition, the data in Figure 3D, E showing ubiquitylation of the endogenous proteins was not convincing. Consequently, the authors should revise this manuscript by providing additional data on the ubiquitylation status of the endogenous proteins.

Second, to support of the proposed mechanism, additional data are needed to show that the problem with Tat transactivation in UBE2O-depleted cells is a failure to recruit P-TEFb to the promoter. This could be addressed in several ways, for example ChIP analysis of the integrated HIV promoter.

Third, the authors should address whether the Tat-UBE20 interaction and ubiquitination of HEXIM1 might have functional consequences for the recently described HEXIM1/NEAT1 lncRNA RNP. Because NEAT1 lncRNA has been implicated in HIV RNA export and replication, this is an intriguing possibility and would add greater and broader significance to the findings described in this paper.

Fourth, does Tat recruitment of CDK9/CCNT1 to the nucleus lead to phosphorylation of multiple substrates that benefit HIV replication? The elevated nuclear level of CDK9 by the Tat-UBE20 interaction may lead to phosphorylation of many substrates, some of which may not be involved in RNAP II transcription. The proposed model implicates UBE2O in controlling the nuclear levels of P-TEFb, and it is important to establish this more convincingly by showing enhanced kinase activity, whether through increased phosphorylation of defined substrates, or increased activity at target genes.

Lastly, it has been previously reported that UBE2A and H2BUb1 are regulated by P-TEFb phosphorylation (Shchebet et al. Cell Cycle 11:2122-2127, 2012). Does UBE2O have a feedback effect on UBE2A and H2B1 ubiquitylation, as would be predicted by the model?

---

## [Author Response]

Essential revisions:The authors identify UBE2O as a cytoplasmic E3 ubiquitin ligase that associates with HIV-1 Tat, and report that it monoubiquitylates the HEXIM1 protein and affects its association with LARP7 and release of P-TEFb to the nucleus, to affect HIV-1 transcription. The data presented are credible but rely on over-expressed proteins and in vitro ubiquitination assays. In addition, the data in Figure 3D, E showing ubiquitylation of the endogenous proteins was not convincing. Consequently, the authors should revise this manuscript by providing additional data on the ubiquitylation status of the endogenous proteins.

We agree that it is important to evaluate the effects of UBE2O on the endogenous proteins, and we have conducted a series of new experiments that do not rely on overexpression. First, we should clarify that Figure 3D does not show data for endogenous HEXIM1 ubiquitination but rather for transfected HEXIM1, and this is now more carefully stated in the last paragraph of the subsection “Tat hijacks UBE2O to mono-ubiquitinate HEXIM1 in a non-degradative manner”. Furthermore, the endogenous ubiquitination data of Figure 3E shows quite conclusively that: 1) HEXIM1 is ubiquitinated and 2) HEXIM1 ubiquitination is completely lost upon UBE2O knockdown, firmly establishing UBE2O as a major endogenous ubiquitin ligase for HEXIM1.

The less pronounced increase in HEXIM1 ubiquitination with Tat expression in Figure 3E can be explained by limiting amounts of endogenous UBE2O. Figure 3—figure supplement 2D demonstrates the typical level of overexpression achieved by UBE2O transfection. Therefore, a lesser degree of Tat-induced endogenous HEXIM1 ubiquitination would be expected from the lower levels of endogenous UBE2O. Further supporting this point is that UBE2O expression alone robustly increases endogenous HEXIM1 ubiquitination (Figure 5B), confirming that UBE2O levels are limiting endogenously.

As requested, we now provide additional data on endogenous protein ubiquitination with Tat expression in Figure 3—figure supplement 3A. First, these new data confirm that HEXIM1, MEPCE, and CCNT1 are all ubiquitinated endogenously. Second, the data demonstrate that Tat increases HEXIM1 and MEPCE ubiquitination, but not the ubiquitination of CCNT1, which matches the substrate specificity of UBE2O from Figure 3A. The text has been updated to incorporate this new data (“Tat expression also increased […]”).

We also provide new tandem affinity purification data (Figure 1F, subsection “UBE2O interacts with the HEXIM1 protein of the 7SK snRNP”), which reveal that Tat, HEXIM1, and UBE2O can form a multi-protein complex. In support of Tat-induced HEXIM1 ubiquitination, we show that endogenous HEXIM1 associated with Tat is in a modified state, compared to HEXIM1 that is purified with LARP7 (Figure 3—figure supplement 3B, subsection “Tat hijacks UBE2O to mono-ubiquitinate HEXIM1 in a non-degradative manner”, last paragraph). This modified species is the size of mono-ubiquitinated HEXIM1. Together, the new data demonstrate the ubiquitination status of the endogenous proteins and confirm that Tat increases the ubiquitination of HEXIM1 in a native context.

Second, to support of the proposed mechanism, additional data are needed to show that the problem with Tat transactivation in UBE2O-depleted cells is a failure to recruit P-TEFb to the promoter. This could be addressed in several ways, for example ChIP analysis of the integrated HIV promoter.

We agree with the comment and now provide evidence showing that UBE2O is essential for P-TEFb recruitment to the HIV-1 promoter (Figure 7E, subsection “Tat releases P-TEFb from cytoplasmic 7SK snRNP complexes for nuclear import and chromatin enrichment”, last paragraph). We performed RNAi-mediated depletion of UBE2O followed by CDK9 ChIP analysis at the HIV-1 promoter. Our results demonstrate that UBE2O knockdown greatly inhibits Tat-dependent enrichment of CDK9 at this DNA sequence, explaining the problem of Tat transactivation in UBE2O-depleted cells.

With the inclusion of the new ChIP result as Figure 7E, the previous data from Figure 7E and F, which presented the DRB-specific effects on P-TEFb localization, have now been moved to a new figure (Figure 8) for presentation purposes.

Third, the authors should address whether the Tat-UBE20 interaction and ubiquitination of HEXIM1 might have functional consequences for the recently described HEXIM1/NEAT1 lncRNA RNP. Because NEAT1 lncRNA has been implicated in HIV RNA export and replication, this is an intriguing possibility and would add greater and broader significance to the findings described in this paper.

This is an interesting point that prompted us to explore the consequences of HEXIM1 ubiquitination on its incorporation into the NEAT1 lncRNP. In particular, we examined how UBE2O knockdown or Tat expression influenced HEXIM1’s interactions with both the 7SK snRNP and NEAT1 lncRNP and have incorporated these results as Figure 6A (subsection “HEXIM1 ubiquitination regulates interactions with the 7SK snRNP and is required for Tat-dependent P-TEFb release from the snRNPP”, second paragraph). As expected, Tat expression decreased the interaction of HEXIM1 with components of the 7SK snRNP, including LARP7, CDK9, and CCNT1. However, Tat expression had no effect on HEXIM1 binding to SFPQ or Ku70 of the NEAT1 RNP. Additionally, UBE2O knockdown increased the association of HEXIM1 with the 7SK snRNP, with no obvious effect on NEAT1 RNP occupancy. This result further supports our model that UBE2O ubiquitination of HEXIM1 leads to P-TEFb release from the inhibitory 7SK snRNP complex. That Tat expression or UBE2O knockdown had negligible effects on HEXIM1-NEAT1 RNP interactions supports the specificity of the UBE2O-dependent HEXIM1 modification for regulating interactions with the 7SK snRNP.

As our new data, which does not rely on UBE2O overexpression, better establishes a connection between UBE2O activity and HEXIM1 interactions with the full 7SK snRNP, we have replaced our previous results from Figure 6A. The previous data demonstrated a loss of only the LARP7 interaction with HEXIM1 upon UBE2O overexpression, which is now supplanted by the more extensive current data with an endogenous HEXIM1 IP and UBE2O RNAi. We have also removed Figure 6—figure supplement 1, which was related to the previous LARP7-specific result of Figure. 6A.

Fourth, does Tat recruitment of CDK9/CCNT1 to the nucleus lead to phosphorylation of multiple substrates that benefit HIV replication? The elevated nuclear level of CDK9 by the Tat-UBE20 interaction may lead to phosphorylation of many substrates, some of which may not be involved in RNAP II transcription. The proposed model implicates UBE2O in controlling the nuclear levels of P-TEFb, and it is important to establish this more convincingly by showing enhanced kinase activity, whether through increased phosphorylation of defined substrates, or increased activity at target genes.

We agree that UBE2O-dependent P-TEFb nuclear import may lead to the increased phosphorylation of multiple substrates. We more closely examined the phosphorylation of RNA polymerase II (RNAP II) at serine 2 (S2) and serine 5 (S5) in the C-terminal domain, as well as the ubiquitination of histone H2B (essential revision 5), which is a known P-TEFb-regulated process. In the case of RNAP II, UBE2O knockdown did not alter S2 phosphorylation, but slightly decreased S5 phosphorylation (Author response image 1). It has recently become clear that RNAP II S5 is a substrate of P-TEFb kinase activity (Czudnochowski et al. 2012, Meyer et al. 2017). In the presence of Tat, RNAP II S2 phosphorylation slightly decreased. Tat has been shown to function as both an activator and repressor at different classes of human genes (Reeder et al., 2015). At down-regulated genes, Tat decreases RNAP II S2 phosphorylation levels, which may explain the bulk decrease in S2-P upon Tat expression. Upon UBE2O knockdown, Tat expression further decreased RNAP II S2-P. A potential explanation is that while Tat can still decrease RNAP II S2-P at down-regulated human genes upon UBE2O depletion, it can no longer maintain normal S2-P levels at activated genes since UBE2O-dependent P-TEFb nuclear import is inhibited. The net effect is a decrease in bulk RNAP II S2-P levels.

Given that Tat expression has a multitude of effects on host cells, it is difficult to be certain why RNAP II S2-P is decreased by Tat in UBE2O knockdown cells. To further explore this mechanism, more extensive studies examining the levels of P-TEFb and RNAP II S2-P would have to be monitored at host genes. We believe this additional level of detail, including the new data on bulk RNAP II phosphorylation, will not further clarify the manuscript and goes beyond the scope of the paper.

Lastly, it has been previously reported that UBE2A and H2BUb1 are regulated by P-TEFb phosphorylation (Shchebet et al. Cell Cycle 11:2122-2127, 2012). Does UBE2O have a feedback effect on UBE2A and H2B1 ubiquitylation, as would be predicted by the model?

Since multiple reports have demonstrated that H2Bub1 is regulated by P-TEFb (Schebet et al. 2012, Pirngruber et al. 2009), we examined how UBE2O knockdown affects H2Bub1 levels. We observed that UBE2O depletion decreases global H2Bub1 levels (see Author response image 2), which is expected from our model. This result with UBE2O knockdown also addresses point 4, showing that UBE2O controls nuclear levels of P-TEFb.

**Author response image 2. respfig2:** 

In addition, we also explored how Tat expression alters H2B ubiquitination. Unexpectedly, Tat decreases global H2B ubiquitination in non-silencing cells, and this effect was inhibited upon UBE2O knockdown. One interpretation is that Tat hijacks P-TEFb for its own transcriptional program, which includes cellular genes (Reeder et al. 2015). This sequestration of P-TEFb limits the amount of kinase available to sustain proper H2Bub1 levels. As UBE2O is critical for Tat-dependent release of P-TEFb from the inhibitory 7SK snRNP, UBE2O knockdown inhibits Tat sequestration of P-TEFb and therefore Tat can no longer decrease global H2Bub1 levels. Altogether, the new data suggest interesting connections between Tat, UBE2O, and H2B ubiquitination. However, the effects of Tat on the system are not easily interpreted or validated without substantial additional experiments that we believe are beyond the scope of the paper, including the preliminary data on H2Bub1.